# MULTIAGENT FINETUNING: SELF IMPROVEMENT WITH DIVERSE REASONING CHAINS

**Vighnesh Subramaniam**[*]
MIT CSAIL
vsub851@mit.edu

**Yilun Du**[*]
Harvard University
ydu@seas.harvard.edu

**Joshua B. Tenenbaum**
MIT CSAIL, BCS, CBMM
jbt@mit.edu

**Antonio Torralba**
MIT CSAIL
torralba@mit.edu

**Shuang Li**[†]
Stanford University
lishuang@stanford.edu

**Igor Mordatch**[†]
UC Berkeley
mordatch@berkeley.edu

## ABSTRACT

Large language models (LLMs) have achieved remarkable performance in recent years but are fundamentally limited by the underlying training data. To improve models beyond the training data, recent works have explored how LLMs can be used to generate synthetic data for autonomous self-improvement. However, successive steps of self-improvement can reach a point of diminishing returns. In this work, we propose a complementary approach towards self-improvement where finetuning is applied to a multiagent society of language models. A group of language models, all starting from the same base model, are independently specialized by updating each one using data generated through multiagent interactions among the models. By training each model on independent sets of data, we illustrate how this approach enables specialization across models and diversification over the set of models. As a result, our overall system is able to preserve diverse reasoning chains and autonomously improve over many more rounds of fine-tuning than single-agent self-improvement methods. We quantitatively illustrate the efficacy of the approach across a wide suite of reasoning tasks.

Project website at https://llm-multiagent-ft.github.io

## 1 INTRODUCTION

Recent breakthroughs in large language models (LLMs) like GPT-3.5 and GPT-4 have demonstrated remarkable proficiency in language generation, comprehension, question answering, and translation (OpenAI, 2023; Touvron et al., 2023). Despite these advancements, LLMs are fundamentally constrained by the data they are trained on, with existing models already using much of the available data on the Internet (Brown et al., 2020). To further enhance the performance of LLMs, recent research on *self-improvement*, where LLMs generate additional synthetic data on which they are trained on (Huang et al., 2022; Yu et al., 2023).

One approach to increase the data available to LLMs is to use powerful existing frontier models like GPT-4 to generate additional supervisory data. However, this approach is limited by the inherent quality of frontier models, preventing models from becoming *better* than the frontier of what the best existing models can accomplish. In addition, such an approach incurs high financial costs due to inference expenses of such large models and is also often legally prohibited with existing commercial-grade models.

An alternative approach is to directly leverage existing language models to generate additional synthetic data for their self-improvement (Zelikman et al., 2022; Bai et al., 2022; Chen et al., 2024b; Yuan et al., 2024). In such works, language models are used to iteratively collect data that they are then finetuned on. However, as models are repeatedly trained, performance gains often plateau relatively quickly as diversity decreases (Figure 1) and the self-improvement loop is often only

---

[*]Equal Contribution, Corresponding authors
[†]Equal advising

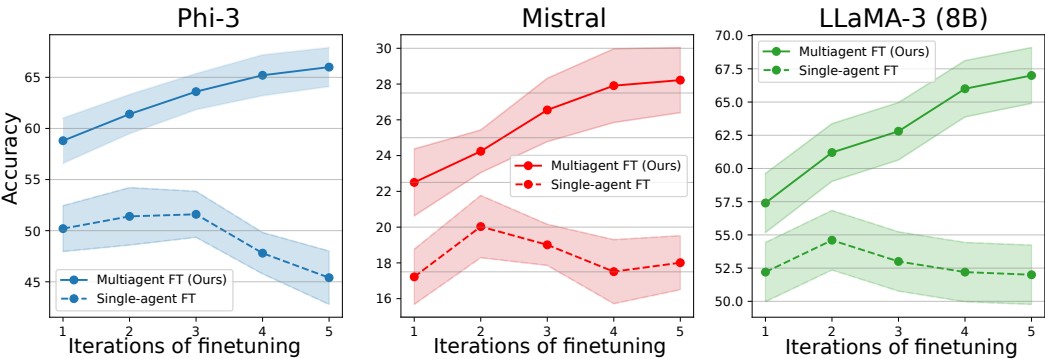

Figure 1: **Multiagent finetuning improves reasoning performance over multiple rounds of finetuning**. Our multiagent finetuning procedure enables models to improve across multiple iterations of finetuing. Results reported on the MATH dataset.

run for two or three rounds (Lu et al., 2023; Song et al., 2024). This limits the applicability of self-improvement to autonomously improve language models, as models can only be improved a limited amount above their base performance.

In this paper, we propose a new approach to self-improvement that can help mitigate the issue of decreased gains of performance after multiple rounds of fine-tuning. Instead of fine-tuning a single model, our method finetunes a multiagent set of language models from the same base model and then independently specializes each model to capture parts of a task of interest. Our key insight is that by finetuning multiple models, we can encourage specialization and diversification across responses, which can enable consistent performance gains over many rounds of fine-tuning. To achieve specialization between models, we fine-tune each model repeatedly on independent subsets of the generated data corresponding to responses from the respective particular model.

Within our multiagent set of models, we propose to specialize models into distinct functionalities within the output generation procedure. First, we specialize a set of models to be generation agents that produce a set of initial responses given queries. Since initial responses can often be suboptimal, especially for challenging reasoning tasks, we further propose to specialize a set of models as critic agents that evaluate and refine the generations of other models. By using this set of distinct models in combination through multiagent debate (Du et al., 2023), we are able to construct a robust feedback loop for generating final responses, with experiments on other multiagent methods in Appendix D.

By training each model on distinct sets of data and roles, our approach fosters specialization across models and promotes diversification within the society of models. Consequently, our system can autonomously improve over many more rounds of finetuning compared to single-agent self-improvement methods (Figure 1). We quantitatively demonstrate the effectiveness of our approach across a comprehensive suite of reasoning tasks, illustrating significant performance gains, as shown in Table 1. In our experiments, we illustrate how our proposed method can be directly applied to both open-source LLMs such as Phi-3, Mistral, and LLaMA-3 as well proprietary LLMs such as GPT-3.5 to substantially improve performance. In addition, the finetuned models can generalize to novel datasets and outperform the baseline methods trained directly on these new datasets.

Overall, our paper has the following contributions: **(1)** We propose to leverage multiagent interaction as an approach to self-improvement with language models. **(2)** We propose to specialize models with distinct roles to enable detailed feedback between agents and to improve the final output quality. **(3)** We quantitatively verify the applicability of our approach across a wide suite of reasoning tasks on both open-source and proprietary language models. **(4)** We demonstrate that the finetuned agents can generalize across different datasets in a zero-shot manner.

## 2 MULTIAGENT FINETUNING OF LANGUAGE MODELS

We provide an overview of our approach towards multiagent finetuning of language models, where we learn a multiagent society of models to accomplish a task. Our method involves two components. We first use a multiagent debate method to construct a finetuning dataset for raining models (though other multiagent generation methods can also be used, see Appendix Section D). We then introduce our approach, multiagent finetuning, where we specialize each LLM model by finetuning each model

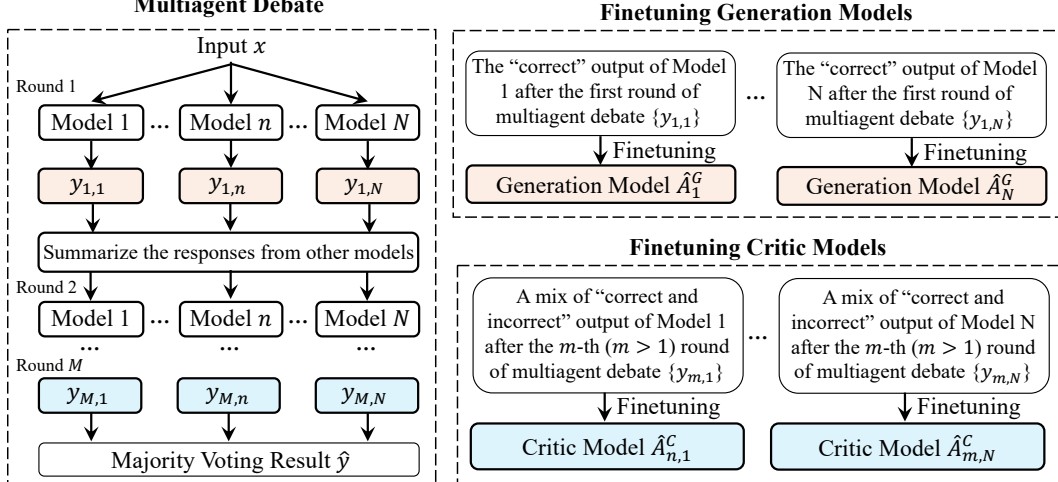

Figure 2: **Overview of Multiagent Finetuning.** We first use multiagent debate and majority voting to create the finetuning datasets (left). These datasets are then used to finetune the generation and critic agents (right). When finetuning generation models, we use the majority voted result ("correct" output) to select first-round responses from each agent. We then finetune critic models using responses from the final round based on whether responses match the majority voted result (mix of "correct and incorrect" outputs). The finetuned models are combined through multiagent debate to generate more accurate answers. In this figure, we illustrate a single finetuning iteration. Applying multiple rounds of finetuning iterations can significantly boost performance.

on its own generated data. An overview of our approach can be seen in Figure 2. We first provide an introduction of our multiagent debate method in Section 2.1. We then discuss how to fine-tune a single model on generated data in Section 2.2, and the proposed multiagent finetuning in Section 2.3 and Section 2.4. We then show how to apply finetuned models for inference in Section 2.5.

## 2.1 MULTIAGENT DEBATE

Multiagent debate (Du et al., 2023) involves a series of $N$ language model agents—either specific copies or finetuned versions of the same model—each tasked with generating a response to a given problem. After the initial responses are generated, a debate round is initiated among the agents. In our paper, we concatenate and summarize the responses from other agents. Each agent is instructed to construct a new response based on its prior response and the summarized responses from the others. The final result is determined by majority vote based on the outputs from the last round of debate. The multiagent debate is illustrated in Figure 2.

## 2.2 FINETUNING MODELS ON GENERATED DATA

We start by considering how to use data generated by multiagent debate data to finetune a single LLM model for self-improvement. Given a set of natural language inputs $\mathcal{D}_{\text{task}} = \{x_i\}$, we use a multiagent debate method (Du et al., 2023), specifically a debate with $N$ agents and $M$ rounds, to generate responses for each input in $\mathcal{D}_{\text{task}}$. We obtain the final predicted output $\hat{y}_i$ for each $x_i$ through majority voting in the last round of debate. We use this to construct a "ground truth" dataset of $\{(x_i, \hat{y}_i)\}$. In the single LLM model setting, we then finetune the model on the set of generated responses $y_i$ which match $\hat{y}_i$ given input $x_i$.

While the final debate results $\hat{y}_i$ are accurate, they often similar in style and methodology. As a result, repeatedly capturing a dataset of $\{(x_i, \hat{y}_i)\}$ pairs for multiple rounds of finetuning often leads to a plateau of self-improvement performance.

## 2.3 FINETUNING MULTIPLE GENERATION AND CRITIC MODELS

Our goal in multiagent finetuning is to create datasets that construct a set of models representing different agents that are diverse and accurately solve problems. Instead of building a single dataset to finetune each model, we propose creating different datasets to finetune different models. A set of

---

**Algorithm 1** Multiagent Finetuning of Language Models

---

**Require:** A pretrained LLM $A$; A set of language inputs $\mathcal{D}_{\text{task}} = \{x_i\}$; The number of agents $N$; The number of debate rounds $M$; The number of finetuning iterations $L$.

1:   $A_1^G, \cdots, A_N^G \leftarrow A$   # Copy the LLM to build $N$ generation agents
2:   $A_1^C, \cdots, A_N^C \leftarrow A$   # Copy the LLM to build $N$ critic agents
3:   # Multiple Iterations of Finetuning
4:   **for** $l = 1 \rightarrow L$ **do**
5:      # Multiagent Debate
6:      **for** $x$ in $\mathcal{D}_{\text{task}}$ **do**   # Iterate over the input tasks
7:         **for** $m$ in $M$ **do**   # M rounds of debate
8:            **if** $m = 0$ **then**
9:               $y_{1,1}, \cdots, y_{1,N} \leftarrow A_1^G(x), \cdots, A_N^G(x)$   # Response of each generation agent
10:           **else**
11:               $x_{m,1}^s, \cdots, x_{m,N}^s \leftarrow$ Summarize the responses from other agents in round $m-1$
12:               $y_{m,1}, \cdots, y_{m,N} \leftarrow A_1^C(x_{m,1}^s), \cdots, A_N^C(x_{m,N}^s)$   # Response of each critic agent
13:           **end if**
14:         **end for**
15:         $\hat{y} \leftarrow$ Majority Voting $\{y_{M,1}, \cdots, y_{M,N}\}$   # Responses of the final round of debate
16:      **end for**
17:      # Multiagent Finetuning
18:      Initialize datasets for finetuning generation models $\{\mathcal{D}_n^G\}_{n=1}^N$
19:      Initialize datasets for finetuning critic models $\{\mathcal{D}_n^C\}_{n=1}^N$
20:      **for** $n$ in $N$ **do**   # Iterate over all the agents
21:         **for** $x$ in $\mathcal{D}_{\text{task}}$ **do**   # Iterate over the input tasks
22:            $\mathcal{D}_n^G \leftarrow \mathcal{D}_n^G \cup \{(x, y_{1,n}) \mid y_{1,n} = \hat{y}\}$   # Add pairs
23:            $\mathcal{D}_n^{C-} \leftarrow \mathcal{D}_n^{C-} \cup \{(x, (y_{1,n}, \cdots, y_{M,n})) \mid y_{1,n} \neq \hat{y}, \, y_{M,n} = \hat{y}\}$   # Add pairs
24:            $\mathcal{D}_n^{C+} \leftarrow \mathcal{D}_n^{C+} \cup \{(x, (y_{1,n}, \cdots, y_{M,n})) \mid y_{1,n} = \hat{y}, \, y_{M,n} = \hat{y}\}$   # Add pairs
25:            $\mathcal{D}_n^C \leftarrow w\mathcal{D}_n^{C-} + (1-w)\mathcal{D}_n^{C+}$   # Combine the datasets
26:         **end for**
27:         $\hat{A}_n^G \leftarrow \text{Finetune}(A_n, \mathcal{D}_n^G)$   # Finetune the generation model
28:         $\hat{A}_n^C \leftarrow \text{Finetune}(A_n, \mathcal{D}_n^C)$   # Finetune the critic model
29:      **end for**
30:      $A_1^G, \cdots, A_N^G \leftarrow \hat{A}_1^G, \cdots, \hat{A}_N^G$   # Generation agent for the next finetuning iteration
31:      $A_1^C, \cdots, A_N^C \leftarrow \hat{A}_1^C, \cdots, \hat{A}_N^C$   # Critic agent for the next finetuning iteration
32: **end for**

---

models are trained as generation agents and others as critic agents. The generation models produce initial responses to input questions. In contrast, the critic models assess the outputs from all generation agents and then select or generate the most effective responses.

**Finetuning Generation Models.** The role of a generation model is to generate accurate responses to input questions. Such models should rely on diverse reasoning chains to promote diversity. Generation agents $A_n^G$ are constructed from the $N$ generation models which generate a response to the given input $x$ (we omit $i$ for simplicity). For each agent, we select its outputs $y_n$ that match the final debate results $\hat{y}$ and construct input-output pairs $(x, y_n)$. The resulting dataset for agent $A_n^G$ is $\mathcal{D}_n^G = \{(x, y_n)\}$. This approach generates a set of finetuning datasets $\{\mathcal{D}_1^G, \cdots, \mathcal{D}_N^G\}$ across all $N$ agents. Each dataset contains different outputs, allowing for specialization and diversification of responses. We finetune each generation model with the corresponding dataset to get $N$ correspondingly finetuned agents $\{\hat{A}_1^G, \cdots, \hat{A}_N^G\}$.

**Finetuning Critic Models.** The role of a critic model is to further provide accurate critiques to responses from other agents and use these responses to provide an updated answer. Simply finetuning generation models isn't sufficient for achieving optimal results, especially for more challenging tasks, due to the lack of a feedback mechanism on their outputs. Critic agents $A_n^C$ are constructed from critic models and evaluate the outputs from all generation agents and then select or synthesize the best responses. This additional step ensures that the system continuously improves and adapts, enhancing overall performance.

In the multiagent debate setting, each agent's output in the last round of debates is represented as $y_{M,n}$, where $M$ denotes the number of debate rounds. We first identify those outputs $y_{M,n}$ that align with the final debate results $\hat{y}$. These consistent outputs, together with the previous responses, are then used to construct input-output pairs $(x, (y_{1,n}, \ldots, y_{M,n}))$ for finetuning the critic models.

To enhance the model's capability to correct incorrect answers generated early in the debate process, we sample a subset of pairs where $y_{1,n}$ differs from $\hat{y}$, but $y_{M,n}$ matches $\hat{y}$ and build a dataset $\mathcal{D}_n^{C-} = \{(x, (y_{1,n}, \ldots, y_{M,n})) | y_{1,n} \neq \hat{y}, y_{M,n} = \hat{y}\}$. This indicates that the answer was successfully corrected by the end of the debates. We also construct another dataset $\mathcal{D}_n^{C+} = \{(x, (y_{1,n}, \ldots, y_{M,n})) | y_{1,n} = \hat{y}, y_{M,n} = \hat{y}\}$ where both $y_{1,n}$ and $y_{M,n}$ match $\hat{y}$, demonstrating the agent's ability to maintain the correct answer throughout the debates. We combine these two datasets to create a comprehensive finetuning dataset for each critic model to construct updated critic agents $A_n^C$:

$$\mathcal{D}_n^C = w\mathcal{D}_n^{C-} + (1 - w)\mathcal{D}_n^{C+}. \tag{1}$$

In the above expression, $w$ is a tunable hyperparameter representing the proportion of data sampled from the first set, while $(1 - w)$ represents the proportion of data sampled from the second set. This method generates a series of datasets $\{\mathcal{D}_1^C, \cdots, \mathcal{D}_N^C\}$ for finetuning the critic models, denoted as $\{\hat{A}_1^C, \cdots, \hat{A}_N^C\}$ after the finetuning process.

## 2.4 Multiple Iterations of Finetuning

The finetuned models are capable of generating responses through multiagent debate. We found that iterative application of the multiagent finetuning allows for continuous learning and adaptation, leading to progressively refined and more accurate responses over time. The finetuned generation agents $\{\hat{A}_1^G, \cdots, \hat{A}_N^G\}$ and critic agents $\{\hat{A}_1^C, \cdots, \hat{A}_N^C\}$ are used to gather datasets for the next iteration through multiagent debate. The algorithm for the proposed approach of $L$ iterations of finetuning is detailed in Algorithm 1. The steps for collecting data for finetuning the generation models are marked in red, and the finetuning of critic models is shown in blue.

## 2.5 Inference

At inference time, we have a set of finetuned generation models which represent generation agents $\{\hat{A}_1^G, \cdots, \hat{A}_N^G\}$, and a set of finetuned critic models which represent critic agents $\{\hat{A}_1^C, \cdots, \hat{A}_N^C\}$. We conduct a multiagent debate among these agents, where each individual generation agent participates in the first round of the debate, followed by each individual critic agent in subsequent rounds. Each agent takes the responses from all other agents and generates a new response in each round of the debate. We found that summarizing the responses from the other agents helps eliminate redundant information while retaining the most important details, thereby further improving performance. The final result is determined by a majority vote based on the responses from the final round of the debate. We provide pseudocode in Algorithm 2.

# 3 Experiments

## 3.1 Language Reasoning Tasks

We evaluate our method and baselines on three language reasoning tasks.

**Arithmetic.** consists of 1,000 generated arithmetic problems in the form $a + b \cdot c + d - e \cdot f$. Following the generation procedure in (Du et al., 2023), each variable is assigned a random value up to a maximum of 30.

**Grade School Math (GSM).** (Cobbe et al., 2021) consists of math word problems that require multi-step mathematical reasoning. Each example includes a problem statement, the numerical answer, and an explanation of the answer.

**MATH.** Hendrycks et al. (2021) consists of competition-level math problems categorized into five difficulty levels. For our experiments, we sample problems from the first three levels.

For each dataset, we randomly select 500 examples for finetuning the language model. Additionally, we select 500 held-out problems for evaluation. We parse the generated answers and evaluate their correctness by comparing them with the ground truth answers. Accuracy is reported based on how

frequently the model returns the correct answer. We also report the standard error of each accuracy value to measure the significance of improvement.

## 3.2 BASELINES

We compare the proposed method with various baselines. In all multiagent settings, we use three agents, and for all debate settings, we conduct two rounds of debates to ensure a fair comparison (additional results with five agents in Appendix Section F).

**Base** utilizes a single language model to process input and generate responses.

**Majority** is a multiagent baseline that selects responses based on a majority vote from multiple agents. If no response secures a majority, one of the potential answers is chosen at random.

**Debate** is a multiagent debate baseline as described in Du et al. (2023). The debate structure is outlined in Figure 2.

**STaR** (Zelikman et al., 2022) iteratively finetunes the language agent using a dataset with ground truth answers for each problem. Initially, the LM generates an answer for each problem, and correct responses, as verified by the ground truth, are added to the finetuning dataset. For problems answered incorrectly, the LM is reprompted with a hint that includes the ground truth answer. Problems where the generated response includes the correct answer are added to the finetuning dataset. The LM is finetuned on the collected dataset. This iterative process of building the dataset and finetuning is repeated until the finetuning loss saturates. The final model is then used for evaluation.

**Majority FT** is a baseline that incorporates both majority voting and finetuning. We prompt the language agents with each problem and conduct a majority vote on their results. We then compile the responses from all agents that align with the majority vote, along with the input, to create a finetuning dataset. The language model is finetuned using this dataset. Finally, we apply majority voting to the outputs of the finetuned model to determine the final answer.

## 3.3 QUANTITATIVE RESULTS

We compare baselines and our method, which was finetuned for only a single iteration ($L = 1$), in Table 1. The accuracy and standard error for each dataset are reported. We use three distinct base language models: three open-source models, Phi-3 4B (Abdin et al., 2024), Mistral 7B (Jiang et al., 2023), and LLaMA-3 8B (Dubey et al., 2024); and one proprietary model, GPT-3.5 (OpenAI, 2022).

Our method outperforms all the baselines. Although "STaR" utilizes ground truth labels for data selection and undergoes multiple iterations of finetuning, it still performs worse than our method, which uses only a single finetuning iteration without access to ground truth. The "Majority", "Debate" and "STaR" methods outperform the "Base" model, demonstrating that majority voting, multiagent debate, and finetuning all contribute to improved performance. "Majority FT" enhances the performance of "Majority" by incorporating a finetuning procedure. Our method is only finetuned on 500 examples and still shows significant improvement over the baselines, particularly on more challenging datasets such as GSM and MATH. Additional evaluations on a larger set of problems and datasets can be found in Appendix Section H.

## 3.4 MULTIPLE ITERATIONS OF FINETUNING

To verify the effectiveness of multiple iterations of finetuning, as described in Section 2.4, we present the performance of our proposed method "Multiagent FT (Ours)" over five iterations of finetuning in Figure 1. We tested this method on two open-source models, Mistral and Phi-3, using the MATH dataset. The results demonstrate that "Multiagent FT (Ours)" consistently improves performance over time. For example, the accuracy of Phi-3 increased from 58.8% to 66.0%, and the accuracy of Mistral improved from 22.5% to 28.2%. Our method with five rounds of finetuning is 12.6% and 9.31% more accurate than the best baseline listed in Table 1 using Phi-3 and Mistral, respectively.

In contrast, finetuning a single agent ("Single-agent FT"), as described in Section 2.2, shows that performance saturates after one iteration of finetuning and starts dropping afterward, indicating potential overfitting to generated responses. This issue occurs when the single model, after several finetuning cycles, becomes fixated on a small range of responses, which limits its diversity and

| LLM | Methods | Arithmetic | GSM | MATH |
|---|---|---|---|---|
| GPT-3.5 (OpenAI, 2022) | Base | $81.99 \pm 0.99$ | $75.60 \pm 1.36$ | $46.83 \pm 2.25$ |
| | Majority | $94.40 \pm 1.03$ | $81.20 \pm 1.24$ | $51.40 \pm 2.23$ |
| | Debate | $98.21 \pm 0.54$ | $83.30 \pm 1.18$ | $55.73 \pm 2.21$ |
| | STaR | $98.38 \pm 0.57$ | $83.60 \pm 1.17$ | $53.00 \pm 2.23$ |
| | Majority FT | $98.40 \pm 0.56$ | $83.70 \pm 1.17$ | $53.40 \pm 2.23$ |
| | Ours | $\mathbf{99.62} \pm 0.28$ | $\mathbf{85.60} \pm 1.11$ | $\mathbf{60.60} \pm 2.18$ |
| Phi-3 (Abdin et al., 2024) | Base | $88.30 \pm 1.44$ | $81.20 \pm 1.74$ | $45.60 \pm 2.10$ |
| | Majority | $91.80 \pm 1.23$ | $81.80 \pm 1.72$ | $47.20 \pm 1.82$ |
| | Debate | $96.20 \pm 0.86$ | $84.40 \pm 1.58$ | $53.40 \pm 2.28$ |
| | STaR | $94.80 \pm 0.99$ | $85.80 \pm 1.21$ | $51.80 \pm 2.06$ |
| | Majority FT | $93.80 \pm 1.08$ | $82.20 \pm 1.71$ | $48.60 \pm 2.16$ |
| | Ours | $\mathbf{99.40} \pm 0.34$ | $\mathbf{88.60} \pm 1.42$ | $\mathbf{58.80} \pm 2.22$ |
| Mistral (Jiang et al., 2023) | Base | $10.80 \pm 0.51$ | $35.60 \pm 1.92$ | $16.60 \pm 1.21$ |
| | Majority | $14.80 \pm 1.17$ | $41.80 \pm 0.88$ | $16.80 \pm 1.25$ |
| | Debate | $19.60 \pm 1.12$ | $52.60 \pm 1.26$ | $18.20 \pm 1.37$ |
| | STaR | $17.40 \pm 0.97$ | $45.50 \pm 1.54$ | $17.84 \pm 1.23$ |
| | Majority FT | $16.40 \pm 0.73$ | $44.60 \pm 1.65$ | $18.91 \pm 1.37$ |
| | Ours | $\mathbf{22.60} \pm 0.97$ | $\mathbf{58.40} \pm 2.11$ | $\mathbf{22.50} \pm 1.87$ |
| LLaMA-3 (Dubey et al., 2024) | Base | $43.20 \pm 2.22$ | $75.00 \pm 1.94$ | $46.80 \pm 2.23$ |
| | Majority | $45.80 \pm 2.23$ | $76.40 \pm 1.90$ | $47.20 \pm 2.23$ |
| | Debate | $48.40 \pm 2.24$ | $78.40 \pm 1.44$ | $51.60 \pm 2.23$ |
| | Majority FT | $49.20 \pm 2.24$ | $77.20 \pm 1.87$ | $52.20 \pm 2.23$ |
| | Ours | $\mathbf{52.00} \pm 2.24$ | $\mathbf{88.60} \pm 1.77$ | $\mathbf{57.40} \pm 2.21$ |

Table 1: **Quantitative results of the proposed method and baselines.** Our method outperforms the baselines across all datasets, as indicated by accuracy (%) $\pm$ standard error. The highest values are highlighted in red, and the second-highest values are highlighted in blue. All results are reported over 500 fixed evaluation problems, expect GSM results for GPT-3.5 which are reported over 1000 fixed evaluation problems (to construct nonoverlapping confidence bars).

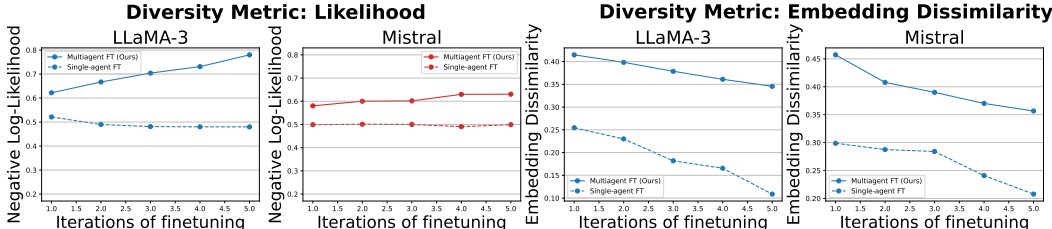

Figure 3: **Diversity is preserved and can improve across iterations of finetuning.** We measure the response diversity of our method and the single-agent finetuning method on the MATH dataset using two diversity measures. The diversity of our method remains consistent over finetuning iterations for one metric and improves for another metric, whereas the diversity of the single-agent method drops significantly.

prevents further enhancement. However, finetuning multiple generation and critic agents using our proposed method increases diversity and consistently improves performance.

## 4 ANALYSIS

In this section, we aim to answer the following questions: 1) How important is the proposed multiagent finetuning procedure? 2) Will it increase response diversity? 3) Can the finetuned agent generalize to other datasets in a zero-shot setting?

### 4.1 ABLATION STUDIES

We examine each component of the proposed method, as shown in Table 2. Multiagent FT (Ours) refers to our proposed method with a single round of finetuning, $L = 1$.

**Multiagent FT w/o summary** removes the summarization step from the multiagent debate. Instead of summarizing, the responses from other agents are directly concatenated and presented to each agent. Summarization helps by eliminating redundant information and retaining the most critical points; therefore, omitting the summarization step can negatively impact performance.

| LLM | Ablations | Arithmetic | GSM | MATH |
|---|---|---|---|---|
| GPT-3.5 (OpenAI, 2022) | Multiagent FT (Ours) | **99.62** ± 0.28 | **85.60** ± 1.67 | **60.60** ± 2.18 |
| | Multiagent FT w/o summary | 99.20 ± 0.40 | 82.20 ± 1.72 | 51.70 ± 2.24 |
| | Multiagent FT w/o critic | 99.20 ± 0.40 | 83.80 ± 1.65 | 50.80 ± 2.24 |
| | Single-agent FT | 99.00 ± 0.45 | 83.60 ± 1.66 | 56.80 ± 2.21 |
| | Single-agent FT w/o debate | 87.20 ± 1.49 | 75.00 ± 1.93 | 48.89 ± 2.23 |
| Phi-3 (Abdin et al., 2024) | Multiagent FT (Ours) | **99.40** ± 0.34 | **88.60** ± 1.42 | **58.80** ± 2.22 |
| | Multiagent FT w/o summary | 98.80 ± 0.51 | 84.40 ± 1.68 | 55.00 ± 2.09 |
| | Multiagent FT w/o critic | 98.20 ± 0.62 | 86.00 ± 1.58 | 56.60 ± 2.22 |
| | Single-agent FT | 97.40 ± 0.71 | 86.80 ± 1.51 | 56.80 ± 2.21 |
| | Single-agent FT w/o debate | 92.20 ± 1.20 | 83.60 ± 1.66 | 50.20 ± 2.24 |
| Mistral (Jiang et al., 2023) | Multiagent FT (Ours) | **22.60** ± 1.87 | **58.40** ± 2.11 | **22.50** ± 1.87 |
| | Multiagent FT w/o summary | 21.80 ± 1.84 | 56.00 ± 1.56 | 20.20 ± 1.55 |
| | Multiagent FT w/o critic | 21.00 ± 1.82 | 54.80 ± 1.60 | 19.01 ± 1.59 |
| | Single-agent FT | 21.20 ± 1.83 | 55.00 ± 2.22 | 19.21 ± 1.69 |
| | Single-agent FT w/o debate | 17.71 ± 1.70 | 51.20 ± 2.24 | 17.22 ± 1.54 |
| LLaMA-3 (Dubey et al., 2024) | Multiagent FT (Ours) | **52.00** ± 2.24 | **88.60** ± 1.77 | **57.40** ± 2.21 |
| | Multiagent FT w/o summary | 50.40 ± 2.24 | 83.20 ± 1.67 | 51.60 ± 2.23 |
| | Multiagent FT w/o critic | 48.60 ± 2.24 | 82.20 ± 1.70 | 50.50 ± 2.23 |
| | Single-agent FT | 48.00 ± 2.23 | 84.40 ± 1.62 | 52.40 ± 2.23 |
| | Single-agent FT w/o debate | 44.00 ± 2.22 | 81.60 ± 1.73 | 48.80 ± 2.24 |

Table 2: **Ablation results**. We examine each component of the proposed method and found that summarization, the combination of critic and generation agents, multiagent finetuning, and multiagent debate all contribute to performance improvement. The accuracy (%) ± standard error is reported.

**Multiagent FT w/o critic**: The critic agents evaluate the outputs from all generation agents and select or synthesize the best responses. Removing the critic agents and only finetuning the $N$ generation agents could hurt performance, as the critic agents play a crucial role of refining the final output.

**Single-agent FT** involves finetuning only a single LLM as covered in Section 2.2 and using it as an agent in multiagent debate. This approach can easily lead to model collapse, where the agent generates similar responses after finetuning, thereby reducing diversity and hurting performance. Therefore, multiagent finetuning is necessary to maintain high performance in reasoning tasks.

**Single-agent FT w/o Debate** further eliminates the debate procedure, with the finetuned LLM generating responses directly. As shown in Du et al. (2023), multiagent debate can significantly boost performance, so removing it could lead to a performance drop.

These results indicate that summarization, the combination of critic and generation agents, multiagent finetuning, and multiagent debate all contribute to performance improvement. Our proposed method integrates these components into a single, unified framework, leveraging their combined benefits.

### 4.2 AGENT RESPONSE DIVERSITY

By finetuning multiple agents with distinct roles, our approach enables us to obtain more diverse responses across rounds of finetuning compared to a single agent. Figure 3 illustrates the diversity of generations from our method and single-agent across rounds of finetuning using two metrics of diversity. We cover one metric of diversity, negative log-likelihood, here and cover the other in Section C.4.

In our first diversity metric, we aim to characterize specialization by tracking the likelihood of responses of other agents using likelihood calculations of a specific agent. If we are increasing diversity, then the log-likelihood of responses from other agents will decrease across iterations of finetuning. The reasoning used by other agents would be considered less common for the specific agent, indicating a divergence in responses. If accuracy increases while likelihood of responses from other agents decreases, this indicates must specialization.

We evaluate the negative log-likelihood (NLL) of responses from other critic agents using another held-out critic agent and plot this over iterations of finetuning. We do the same with Single-Agent FT, using responses from other agents and evaluate likelihood using a held-out agent. Larger NLL values indicate that the model has assigned low likelihood to a sequence and lower NLL values indicate that the model has assigned higher likelihood to a sequence. We measure this over iterations of finetuning for our method as well as Single-Agent FT.

We compute the diversity across all test examples and present the results in Figure 3. For the "Single-agent FT", all agents are the same finetuned language models, and $M = 1$. We notice that NLL increases across iterations of finetuning for our method, meaning that responses from other critic agents are more diversity according to our held-out critic agent. Moreover, our responses are more diverse than using Single-Agent FT. This aligns with our previous observation that diverse responses can mitigate model collapse and prevent the model from overfitting to the finetuning data, leading to better performance. We also include another metric, embedding dissimilarity, as a further comparison, finding that responses from our method preserves diversity, where as diversity reduces significantly with Single-agent FT. We provide additional metrics for evaluating diversity in generations in Appendix Section C, and similarly find that multiagent finetuning preserves the final diversity of generations.

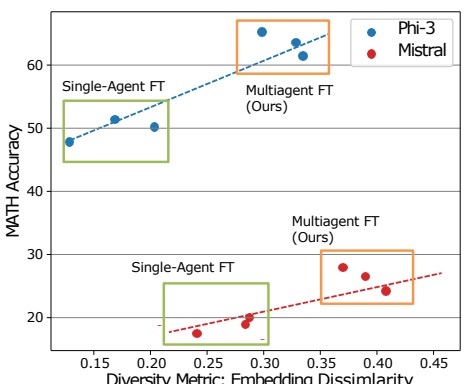

Figure 4: **Relationship between accuracy and diversity.** We visualize the relationship between embedding dissimilarity and MATH accuracy across rounds of finetuning. Our multiagent finetuning preserves diversity across rounds of finetuning while improving accuracy.

We further analyze the relationship between diversity and performance and show this in Figure 4. Specifically, we see that an improvement in the diversity of responses correlates positively with an improvement in performance across rounds of finetuning across both Phi-3 and Mistral models. This suggests that in general, increasing the diversity of responses can be helpful for improvement over multiple rounds of fine-tuning. In Appendix Section E, we compare our approach with additional approaches to improve the diversity of samples such as increasing the temperature at which samples are generated, or using unique IDs in a single language to simulate a single agent. We find that our approach outperforms these baselines.

## 4.3 ZERO-SHOT GENERALIZATION

We investigate the zero-shot generalization of the proposed method across different datasets. Specifically, we use generation and critic agents finetuned on the MATH dataset and evaluate their performance on 100 randomly sampled examples from the GSM dataset. We compare our method to baseline methods used in Table 1. These baselines are trained on the GSM dataset. All methods use Mistral as the base LLM. Figure 5 shows that our method surpasses all the baseline methods, even though it has never seen data from the GSM dataset, indicating the strong zero-shot generalization capability of the proposed method. We show further results in Section H.3.

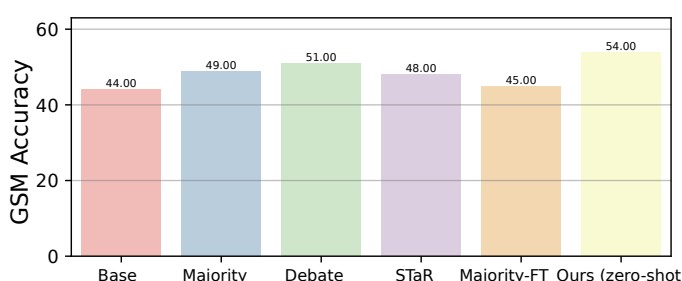

Figure 5: **Zero-shot generalization of the proposed method**. Our method demonstrates zero-shot generalization capabilities. When trained on the MATH dataset, it can effectively generalize to the GSM dataset. It outperforms all the baselines that are trained on the GSM dataset.

## 5 RELATED WORK

Finetuning methods generally fall into three categories: human-in-the-loop, distillation, and self-improvement. We briefly cover the first two categories and spend more time on self-improvement, which is more related to our work.

**Finetuning with human-in-the-loop and distillation**: Several human-in-the-loop methods have been introduced for finetuning, most noticeably RLHF (Christiano et al., 2017; Sun et al., 2023) and

DPO (Rafailov et al., 2024). These methods have been employed as part of *instruction tuning* (Zhang et al., 2023), improving the generated responses to instructions. Several instruction tuning datasets (Wang et al., 2022; Longpre et al., 2023) have been released publicly, some with human-generated responses. Other datasets have been constructed using the second category of finetuning methods, distillation, whereby a much larger, highly performant LLM is used to generate data that finetunes a smaller LLM (Peng et al., 2023; Liu et al., 2024). These approaches have been used to build recent LLMs such as Alpaca (Taori et al., 2023) or Vicuna (Chiang et al., 2023) using responses generated by GPT-3.5 or GPT-4 (Achiam et al., 2023).

**Finetuning with self-improvement**: Self-improvement methods (Huang et al., 2022; Yu et al., 2023; Yuan et al., 2024; Hsieh et al., 2023; Welleck et al., 2022) improve the performance of LLMs through the finetuning. Common approaches include iterated learning (Anthony et al., 2017; Vani et al.; Polu et al., 2022; Xu et al., 2024) where solution/methods discovered by optimization on prior data are used to uncover further solutions or, in this context, provide additional finetuning data. Some of the main papers we use for comparison finetune using bootstrapping through rationale generation (Zelikman et al., 2022; Lee et al., 2024; Pang et al., 2024; Zhang et al., 2024; Lu et al., 2023) or use self-play/self-training methods through reinforcement learning (Chen et al., 2024b; Yuan et al., 2024; Chen et al., 2024a). Most methods find that using self-generated rationales leads to significant improvement when finetuning. However, these works and many others rely on access to ground truth answer. Overall, existing works often show a plateauing effect with limited boosts in improvement after several rounds of fine-tuning. Our work proposes to use multiagent interaction as an approach to get more consistent gains after multiple rounds of finetuning.

**Multiagent Interaction**: Our work builds on the combination of finetuning and multiagent interaction systems. We primarily incorporate multiagent debate (Du et al., 2023; Chan et al., 2023; Pham et al., 2023; Liang et al., 2023) due to its success in improving factuality and reasoning in LLMs in a variety of tasks at inference time. Several other multiagent interactions could also serve as the basis for this paper. Tree-of-thought (Yao et al., 2024; Long, 2023) and graph-of-thought (Besta et al., 2024) represent two common multiagent interaction systems over LLMs that incorporate responses across multiple LLMs, which improves reasoning. Other works (Wu et al., 2023) have designed more flexible systems for multiagent conversations built on structured program synthesis rather than natural language. Prior work has also focused on incorporating multiagent interaction into domains beyond factuality and reasoning such as strategy and communication games (Abdelnabi et al., 2023). More recently, this has led to multiagent interaction systems over LLMs that have optimized via equilibrium search for factuality and reasoning tasks (Jacob et al., 2023b;a). In contrast to existing works, our work aims to use multiagent interaction as a method to finetune language models.

## 6    CONCLUSION AND LIMITATIONS

**Limitations.** In comparison to existing works in single model finetuning, multiagent finetuning is substantially more expensive at both training and inference time as multiple copies of a model need to be trained and run. To run multiagent finetuning experiments on open source models, we used either four H100 GPUs or four A100 GPUs. Models took between 120GB - 240GB of GPU memory and inference took between 12-24 hours across multiple GPUs. To improve the training time of multiagent models, it may be interesting to instead share weights across different instances of models. To improve inference time in multiagent models, we can directly distill the debate procedure into a single modelor use quantization as part of finetuning.

**Conclusion.** In this paper, we have introduced a novel multiagent finetuning framework that significantly enhances the performance and diversity of language models. By employing a society of agents with distinct roles, our method effectively improves the feedback mechanism and overall output quality, mitigating the limitations inherent in single-agent self-improvement methods. This system allows for autonomous self-improvement through iterative finetuning, leading to substantial performance gains across a comprehensive suite of reasoning tasks. Importantly, our approach is versatile and can be applied to both open-source and proprietary LLMs, ensuring broad utility and impact. Additionally, our method can be integrated with other finetuning approaches such that incorporate human feedback such as RLHF or DPO, which we leave to future work. This work opens new avenues for future research in language model enhancement and sets a foundation for further advancements in the field.

ACKNOWLEDGMENTS

This work was supported by the Center for Brains, Minds, and Machines, NSF STC award CCF-1231216, the NSF award 2124052, the MIT CSAIL Machine Learning Applications Initiative, the MIT-IBM Watson AI Lab, the CBMM-Siemens Graduate Fellowship, the DARPA Mathematics for the DIscovery of ALgorithms and Architectures (DIAL) program, the DARPA Knowledge Management at Scale and Speed (KMASS) program, the DARPA Machine Common Sense (MCS) program, the Air Force Office of Scientific Research (AFOSR) under award number FA9550-21-1-0399, the United States Air Force Research Laboratory and the Department of the Air Force Artificial Intelligence Accelerator under Cooperative Agreement Number FA8750-19-2-1000, and ONR MURI grant N00014-22-1-2740. The views and conclusions contained in this document are those of the authors and should not be interpreted as representing the official policies, either expressed or implied, of the Department of the Air Force or the U.S. Government. The U.S. Government is authorized to reproduce and distribute reprints for Government purposes notwithstanding any copyright notation herein.

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

## A  APPENDIX SUMMARY

We add additional details for our methods and experiments as well as additional results to provide more evidence of improvements with multiagent finetuning. In Section B, we provide additional details on summarization, inference and training details using multiagent finetuning with debate. In Section C, we cover additional metrics for measuring diversity in agent responses based (1) consensus and (2) KL-divergence (3) likelihood. Both metrics show that diversity is maintained or increases while accuracy increase over rounds of finetuning. In Section D, we introduce a cooperative approach for composing agent responses rather than a competitive approach through multiagent debate. We apply multiagent finetuning with the cooperative approach to analyze whether our method is agnostic to the approach style. We find strong similar improvements when our method is applied to a cooperative approach. In Section E, we include an additional baseline based on Single Agent FT where we increase the sampling temperature applied across all agents. This is a proxy for increasing diversity that is complementary to our method. We find that multiagent finetuning significantly outperforms methods that modify temperature to artificially induce diversity. In Section F, we add an additional experiment where we apply multiagent finetuning to responses across 5 agents instead of 3. We see significant improvements in performance when using additional agents. In Section G, we present a simple mathematical model illustrating how multiagent finetuning can improve diversity. Finally, in Section H, we present additional evaluations of multiagent finetuning across a wide suite of datasets.

## B  METHODOLOGY DETAILS

### B.1  SUMMARIZATION DETAILS

As done in Du et al. (2023), we incorporate summarization into the multiagent debate procedure. In summarization, we have an LLM agent take responses from other agents as input and summarize the answers to the responses. During round $m$ of debate, we introduce a summarization agent $A_n^S$ which takes responses from the other $N-1$ agents in the last round, $(y_1^{m-1}, \cdots, y_{n-1}^{m-1}, y_{n+1}^{m-1}, \cdots y_N^{m-1})$ and generates a summary of the responses $x_{m,n}^s$. This summary is sent to the critic agent $A_n^C$ to generate a new response.

### B.2  INFERENCE DETAILS

The pseudocode of our method for inference is shown below. .

---

**Algorithm 2** Inference

---

**Require:** A set of finetuned generation agents $\{\hat{A}_1^G, \cdots, \hat{A}_N^G\}$; A set of finetuned critic agents $\{\hat{A}_1^C, \cdots, \hat{A}_N^C\}$;
  A test set of language inputs and ground truth responses $\mathcal{D}_{\text{task}} = \{x_i, y_i\}$; The number of agents $N$; The
  number of debate rounds $M$.
1: $\text{success} \leftarrow 0$
2: **for** $x, y$ in $\mathcal{D}_{\text{task}}$ **do**   # Iterate over the input tasks
3: **for** $m$ in $M$ **do**   # M rounds of debate
4: **if** $m = 0$ **then**
5: $y_{1,1}, \cdots y_{1,N} \leftarrow \hat{A}_1^G(x), \cdots, \hat{A}_N^G(x)$   # Response of each generation agent
6: **else**
7: $x_{m,1}^s, \cdots, x_{m,N}^s \leftarrow$ Summarize the responses from other generator agents
8: $y_{m,1}, \cdots, y_{m,N} \leftarrow \hat{A}_1^C(x_{m,1}^s), \cdots, \hat{A}_N^C(x_{m,N}^s)$   # Response of each critic agent
9: **end if**
10: **end for**
11: $\hat{y} \leftarrow$ Majority Voting $\{y_{M,1}, \cdots, y_{M,N}\}$   # Responses of the final round of debate
12: $\text{success} \leftarrow \text{success} + \mathbb{I}(\hat{y} = y)$
13: **end for**
14: $\text{Accuracy} \leftarrow \frac{\text{success}}{|\mathcal{D}|}$

---

### B.3 Experimental Details

For all open-source models, we perform finetuning using a total of eight 40GB A100 GPUs and four 80GB H100 GPUs. The evaluation of individual inference times for multi-agent finetuning with open-source models took approximately 30 to 36 hours.

**Phi-3** We ran our results using `Phi-3-Mini-128K-Instruct` which has 4 billion tunable parameters. We finetune the entire model end-to-end (no LoRA or memory adaptation) on two 40GB A100 GPUs or one 80GB H100 GPU and run a total of two epochs of finetuning for generation agents and one epoch of finetuning for critic agents. We use a batch size of 1 and a learning rate of $5e^{-6}$ for generation agents and $5e^{-7}$ for critic agents. When applying multiple iterations of finetuning, we use a learning rate of $5e^{-7}$, and a weight decay of $1e^{-3}$ across both generation and critic agents. Models are finetuned with a fixed training set of 500 randomly selected questions (where we do not provide answer annotations for the questions) and then evaluated on a separate test set of 500 randomly selected questions.

**Mistral** We ran our results using `Mistral-7B-Instruct-v0.2`, which has 7 billion tunable parameters. We finetune the entire model end-to-end (no LoRA or memory adaptation) on four 40GB A100 GPUs or two 80GB H100 GPUs and run a total of one epoch of finetuning. We use a batch size of 1 and a learning rate of $5e^{-7}$ for generation agents and $5e^{-7}$ for critic agents and a weight decay of $1e^{-2}$. When applying multiple iterations of finetuning, we use a learning rate of $5e^{-7}$ across both generation and critic agents. Models are finetuned with a fixed training set of 500 randomly selected questions (where we do not provide answer annotations for the questions) and then evaluated on a separate test set of 500 randomly selected questions.

**LLaMA-3** We ran our using `Meta-Llama-3-8B-Instruct`, which has 8 billion tunable parameters. We finetune the entire model end-to-end (no LoRA or memory adaptation) on three 80GB H100 GPUs and run a total of two epochs of finetuning. We use a batch size of 1 and a learning rate of $5e^{-7}$ for generation agents and $2e^{-7}$ for critic agents. When applying multiple iterations of finetuning, we use a learning rate of $5e^{-7}$ across both generation and critic agents as well as a weight decay of $1e^{-2}$. Models are finetuned with a fixed training set of 500 randomly selected questions (where we do not provide answer annotations for the questions) and then evaluated on a separate test set of 500 randomly selected questions.

**GPT-3.5** We ran our results on the `gpt-3.5-turbo-0613` model. We use the finetuning API and run a total of two epochs of finetuning, using a batch size of 1 and a learning rate multiplier of 1. Models are finetuned with a fixed training set of 500 randomly selected questions (where we do not provide answer annotations for the questions) and then evaluated on a separate test set of 500 randomly selected questions.

## C Diversity Metrics

We cover different metrics for measuring diversity for both Phi-3 and Mistral to provide an overview of the diversity of our method in comparison to Single Agent FT.

### C.1 Consensus

We further analyze the diversity of responses from our method to show that diversity is preserved. Rather than using text embeddings, we further measure the *consensus* among agents as a more interpretable alternative. This is measured as the proportion of agents that have the same final answer in a given round of debate. We take an average of this proportion across all 500 problems used for evaluation. To obtain the mean consensus of our single agent finetuning baseline, we prompt the single-agent finetuned model 3 times, take a majority vote over generated answers, and find the proportion of agents that had a generated answer that was the majority vote. In order to convert this to diversity, we take the difference of the mean consensus value from 1, which represents the average number of agents with a different response from the consensus answer.

We measure diversity as the inverse of consensus. Specifically, we consider the agent responses in the final round of debate $\{y_{M,1}, \cdots, y_{M,N}\}$ that match the majority-voted final response $\hat{y}$. The

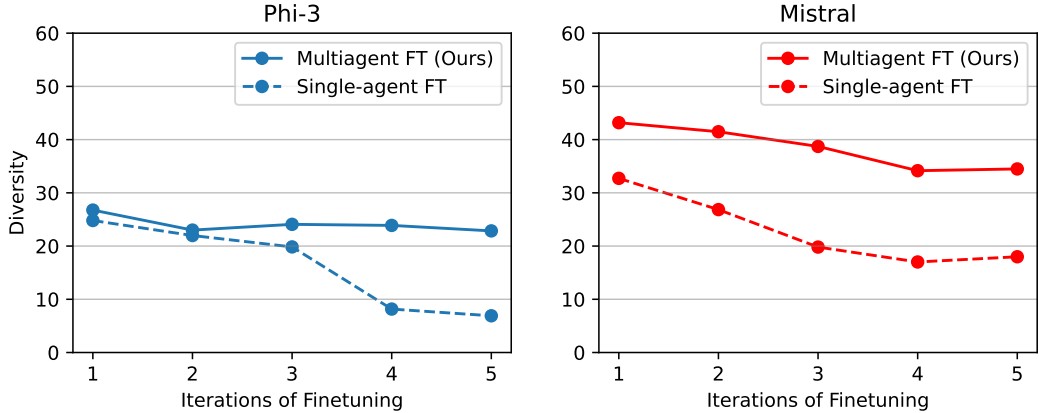

Figure 6: **Consensus: Response diversity across finetuning iterations**. We measure the response diversity based on agent consensus of our method and the single-agent finetuning method on the MATH dataset. The diversity of our method remains consistent over finetuning iterations, whereas the diversity of the single-agent method drops significantly.

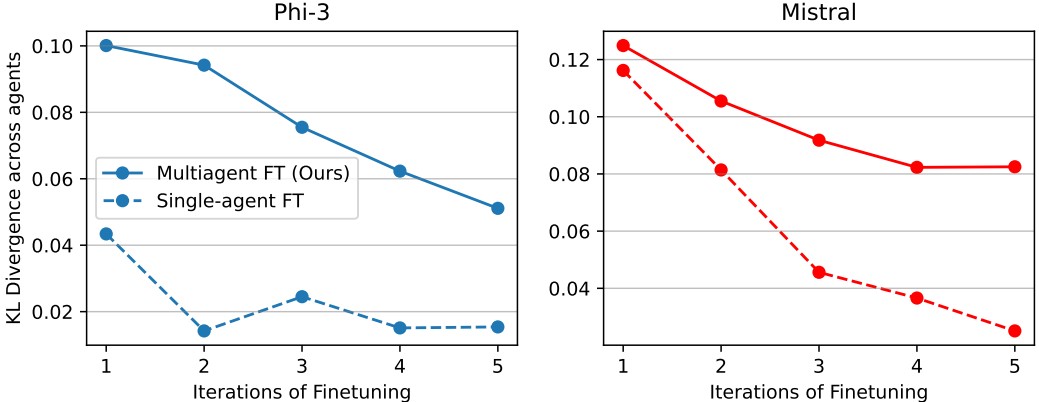

Figure 7: **KL-Divergence: Response diversity across finetuning iterations**. We measure diversity based on the KL-divergence between the probabilities of the output tokens between agents. Similar to our likelihood measurement, we find that diversity is preserved across rounds of finetuning.

consensus is computed as the percentage of responses in $\{y_{M,1}, \cdots, y_{M,N}\}$ that match $\hat{y}$:

$$\text{Consensus} = \frac{\sum_{n=1}^{N} \mathbb{I}(y_{M,n} = \hat{y})}{N},$$

where $\mathbb{I}$ is the indicator function. Diversity is then given by Diversity $= 1 - $ Consensus.

We show results in Figure 6. As seen with our prior metric, embedding dissimilarity, we can preserve diversity based on the responses given by the agents, rather than based on the embeddings of a language model.

## C.2 KL-DIVERGENCE

We next measure diversity by computing KL divergence between the probability distributions computed based on the final answers from different agents. We estimate the probability distribution of each agent's response using the likelihoods from Gemma-2 (2B) For each test example, we compute the KL divergence between the responses of any two agents and then average the values from all pairs of agents to determine the overall KL divergence.

We see results in Figure 7. Specifically, we see that diversity is preserved using our method whereby KL-divergence is consistently higher than the single-agent finetuning baseline.

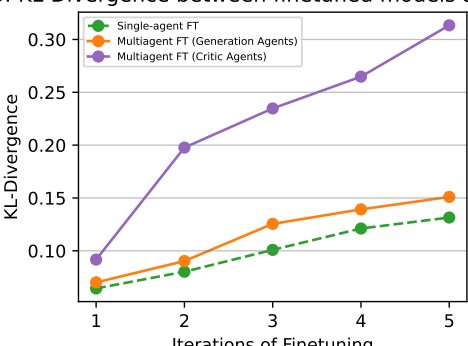

Figure 8: **KL Diversity between finetuned and unfinetuned LLM.** We measure the KL-divergence between likelihoods of responses from finetuned agents and base LLM agents for single-agent finetuning and generation/critic agents from multiagent finetuning. Likelihoods are calculated using Gemma-2 (2B). We find that our method diverges from the base LLM probabilities and furthermore, critic agents have better divergence in responses and our method has better diversity metrics than single-agent FT.

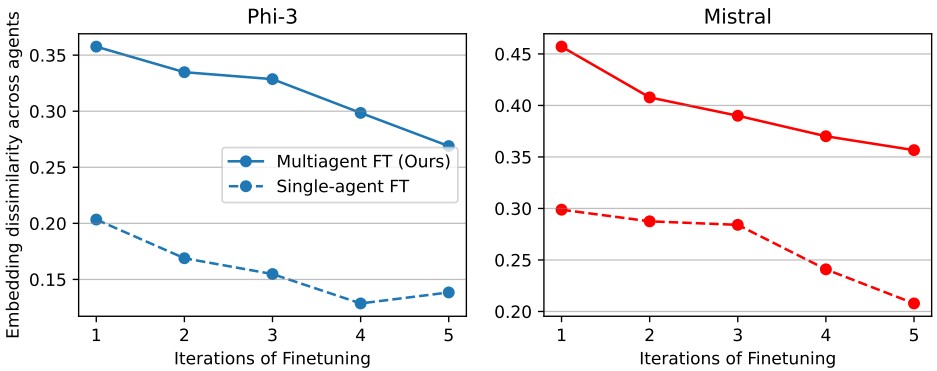

Figure 9: **Embedding Dissimilarity: Response diversity across finetuning iterations.** We measure the response diversity based on the embedding dissimilarity between the responses of different agents, where embeddings are computed using the T5-3B encoder. We notice that similar to likelihood measurement, we find that diversity is preserved across rounds of finetuning.

## C.3 KL-DIVERGENCE ACROSS MODELS

We further analyze diversity by comparing the KL-divergence of generation and critic agents with the likelihood of responses from the base LLM model across iterations of finetuning.

We measure the KL-divergence between each agent responses and responses from a base LLM for 500 MATH examples. We average KL-divergence across all examples for each iteration of finetuning. We apply this measure to agents formed through Single Agent-FT and to generation and critic agents formed through our method. For Single-Agent FT, we find the KL divergence for each finetuned agent and average the KL-divergence across all examples and all agents per iteration of finetuning. For our method, we separate generation and critic agents and find the average KL-divergence for both. We measure likelihoods using Gemma-2 (2B), similar to Figure 7.

We show results in Figure 8. We see that critic agents generally have higher KL-divergences from the base LLM and both critic and generation agents have higher KL-divergences across iterations of finetuning.

## C.4 EMBEDDING DISSIMILARITY

Finally, we analyze diversity by measuring the embedding dissimilarity between responses of different agents.

Specifically, we consider agent responses in the final round of debate $\{y_{M,1}, \cdots, y_{M,N}\}$ that match the majority-voted final majority-voted final response $\hat{y}$. For each response, we obtain pretrained

| LLM | Methods | Arithmetic | GSM | MATH |
|---|---|---|---|---|
| GPT-3.5 | Cooperative (Base) | 96.60 ± 0.83 | 81.80 ± 1.73 | 53.60 ± 2.23 |
| | Cooperative (FT) | **98.80** ± 0.39 | **84.00** ± 1.64 | **56.40** ± 2.21 |

Table 3: **Cooperative Finetuning.** Our method supports fine-tuning in cooperative settings, where agents work together (e.g., 3 agents, 2 rounds).

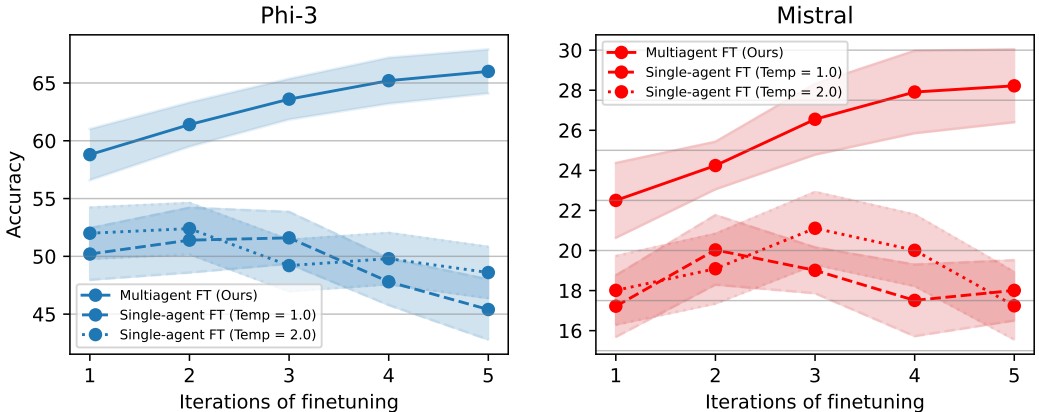

Figure 10: **Inducing diversity through increasing temperature**. We introduce an additional baseline where we apply the Single-Agent FT baselin with a temperature of 2. By increasing the sampling temperature, we allow the model to generate more diverse responses. We observe that our method out-performs higher temperature settings, which demonstrates that temperature does not increase diversity in a way that is useful for accuracy.

contextual word embeddings from a held-out language model, in this case the T5-3B encoder model (Raffel et al., 2020).

We feed each agent response to the T5 encoder model to obtain word embeddings and extract the embedding associated with the classification token `[CLS]`. As done in prior work, we use this embedding as a representation of the sequence. We compare the similarity of the agent responses using cosine similarity of the `[CLS]` embeddings. Since cosine similarity measures similarity, to obtain a metric for diversity, we take the complement of cosine similarity by subtracting the value from 1.

## D  COOPERATIVE FINETUNING

In this paper, our method mainly builds on a competitive approach for composing agent responses with multiagent debate. Our approach for multiagent finetuning can be applied to both the competitive setting, where critic agents provide feedback to generator agents, and cooperative settings, where agents work together in a "mixture of experts" style to generate answers. Instead of prompting agents to critique responses from other agents, in the second round of conversation, we prompt agents to cooperate with other agents. We ask each agent to generate a new response by merging their own response with the responses of other agents, using the prompt "Can you derive a new solution by combining your solution with the solutions of other agents?". Under this cooperative setting, the proposed multi-agent finetuning improves the performance, as demonstrated by Cooperative (FT) outperforming Cooperative (Base).

We show results in Table 3. More specifically, we see that we can finetune with a cooperative method with multiagent finetuning and achieve similar improvements in performance. This demonstrates that our method can be applied to other multiagent prompt settings as a general finetuning method for LLMs.

## E  ADDITIONAL COMPARISONS

We compare our approach to two additional approaches to improve the diversity of reasoning chains.

| LLM | Methods | Arithmetic | GSM | MATH |
|---|---|---|---|---|
| GPT-3.5 | Debate | $99.40 \pm 0.34$ | $85.40 \pm 1.58$ | $58.20 \pm 2.22$ |
| | Majority FT | $99.60 \pm 0.28$ | $86.20 \pm 1.54$ | $59.00 \pm 2.19$ |
| | Ours | $\textbf{100.00} \pm 0.00$ | $\textbf{88.20} \pm 1.44$ | $\textbf{62.80} \pm 2.16$ |
| Phi-3 | Debate | $97.40 \pm 0.71$ | $86.00 \pm 1.55$ | $55.20 \pm 2.22$ |
| | Majority FT | $95.80 \pm 0.90$ | $84.80 \pm 1.61$ | $53.20 \pm 2.23$ |
| | (Ours) | $\textbf{99.80} \pm 0.20$ | $\textbf{89.40} \pm 1.38$ | $\textbf{60.40} \pm 2.19$ |

Table 4: **More agents of debate**. With 5 agents and 2 rounds of debate, our methods still outperform the baselines and show better results than the 3 agents and 2 rounds of debate results presented in Table 1 of the main paper.

| LLM | Methods | MATH |
|---|---|---|
| LLaMA-3 (Dubey et al., 2024) | Base | $46.80 \pm 2.23$ |
| | Debate | $51.60 \pm 2.23$ |
| | Unique ID | $50.80 \pm 2.24$ |
| | Ours | $\textbf{57.40} \pm 2.21$ |

Table 5: **Unique ID vs Multiagent Finetuning.** We introduce an additional comparison to multiagent finetuning where we feed a unique ID token to each agent, corresponding to a generation or critic agent. We find that this is not comparable to improvements on multiagent finetuning.

## E.1 MODULATING TEMPERATURES

We first consider inducing diverse responses from LLM agents by increasing the temperature of generation. We add an additional baseline where we vary the temperature of agents finetuned using Single Agent-FT. Higher temperature values may be a proxy for more diverse responses. We show results over rounds of finetuning in Figure 10.

We see that our method surpasses the performance of this baseline. This likely because higher temperature values can reduce accuracy due to increased variability of samples. Our method preserves diversity of responses while increasing accuracy using a more carefully designed finetuning method.

## E.2 UNIQUE ID FOR AGENTS

We next considering an additional comparison to multiagent finetuning that can preserve diversity while reducing the cost of finetuning. The method involves using a unique identifier as part of the prompt fed to each agent. We feed each generation agent an ID given by `GEN1`, `GEN2`, etc. Similarly, each critic agent is given an ID `CRIT1`, `CRIT2`, etc. Additionally, we provide a short description to the agent, explaining what the ID refers to. For generation agents, we state that the agent is tasked with creating a solution. For critic agents, we state that the agent is tasked with evaluating and improving responses. The ID is presented to the agent at the beginning of each prompt, marked by the string `Agent ID: GEN1 (This is a generation agent tasked with creating a solution.)` as an example of the ID fed to generation agent 1.

We compare the unique ID approach on the same 500 MATH examples reported in Table 1. Results are shown in Table 5. We find that multiagent finetuning performs significantly better and that using unique IDs is fairly similar to debate. This demonstrates that mechanisms for generating solutions and critiquing them is unlocked via finetuning.

## F ADDITIONAL AGENTS IN DEBATE

In Table 4, we show the influence additional agents with finetuning. We use 5 agents and 2 rounds of debate. We find that additional agents improves results as noted in prior work (Du et al., 2023) over 3 agents, 2 rounds of debate. This also implies that our method will scale with larger number of finetuned agents.

| LLM | Methods | MATH |
|---|---|---|
| LLaMA-3 (Dubey et al., 2024) | Base | $24.40 \pm 1.92$ |
| | Majority | $25.20 \pm 1.94$ |
| | Debate | $29.80 \pm 2.05$ |
| | Majority FT | $28.00 \pm 2.01$ |
| | Ours | $\mathbf{34.20} \pm 2.12$ |

Table 6: **Additional Evaluation of Multiagent Finetuning on more difficult tasks.** Our method outperforms the baselines on more difficult tasks including examples from all levels of MATH. This shows the applicability of our method in more broad settings.

## G   MATHEMATICAL MODEL OF DIVERSITY OVER ROUNDS OF FINETUNING

We consider a simple mathematical model illustrating how diversity can arise by finetuning models only on answers that they are accurate on. Consider a training dataset of problems in three topics, A, B, and C as well as three models we train all initialized from the same base model. For each model, we assign a specialization skill score $S_A$, $S_B$, $S_C$ between 0 and 1, representing how accurate the model is at answering questions in the specified topic. All three models are initialized to have a skill of 0.33 on each topic. The specialization $S_i$ for each topic $i$ corresponds to the percentage of questions in topic $i$ the model get accurate, where $S_A$ of 0 represents that a model would get 0% of questions in topic A correct.

At each iteration, a model is trained on all questions it answers correctly in each topic. This increases the specialization skill score by fraction of training the model saw for each specific topic. Formally, the updated skill of model $A$ at iteration $t$ would be:

$$S_A^t = S_A^{t-1} \left( 1 + \frac{S_A^{t-1}}{S_A^{t-1} + S_B^{t-1} + S_C^{t-1}} \right). \tag{2}$$

To account for a finite amount of capacity in each model, after the above skill update, the skills across all models at iteration $t$ are then normalized to have a sum of one. Without loss of generality, assume that at iteration t, $S_A^t$ is larger than $S_B^t$ and $S_C^t$ (which happens by random chance, since we have a finite number of questions). Under the update rule described, the ratio $S_A^{t+1}$ to $S_A^t$ is given by

$$\left( 1 + \frac{S_A^t}{S_A^t + S_B^t + S_C^t} \right) / \left( \sum_{i \in \{A,B,C\}} \left( 1 + \frac{S_i^t}{S_A^t + S_B^t + S_C^t} \right) S_i^t \right). \tag{3}$$

Since $S_A^t$ is greater than or equal to $S_i^t$, the above expression is greater than or equal to

$$\left( 1 + \frac{S_A^t}{S_A^t + S_B^t + S_C^t} \right) / \left( \sum_{i \in \{A,B,C\}} \left( 1 + \frac{S_A^t}{S_A^t + S_B^t + S_C^t} \right) S_i^t \right) = 1, \tag{4}$$

where we use the identity that the sum of $S_i^t$ is equal to 1 to indicate since they are normalization of the scores. We thus have that $S_A^{t+1}$ will be larger than $S_A^t$, with specialization on topic A monotonically increasing over iterations of training.

Since a priori the model has no preference for any particular topic, random sampling each initial base model will lead to skill preference over a different random topic. This repeated procedure will then eventually result in models specializing in either topic A, B, C, ensuring diversity across models. This mathematical model is similar to the multiagent finetuning procedure in the paper, where we selectively train generators and critics on datasets they are accurate on and illustrate how they can then specialize on different portions of data.

## H   ADDITIONAL EVALUATIONS

### H.1   LARGER MATH EVALUATION

To further evaluate multiagent finetuning, we evaluate on the MATH dataset across all 5 levels of difficulty, instead of selecting examples from levels 1-3. We extract 500 examples for training and 500 examples for testing and evaluate on LLaMA-3.

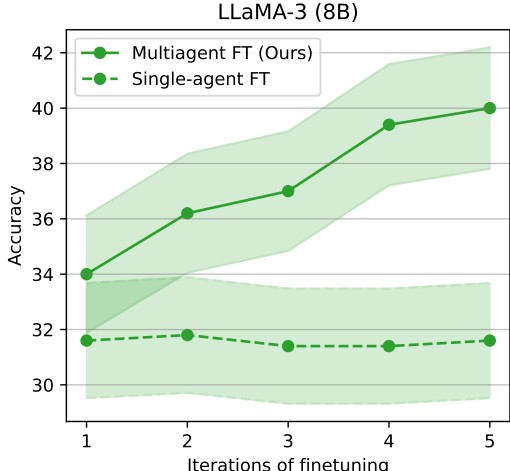

Figure 11: **Multiple iterations of finetuning over all levels of MATH**. We apply multiple iterations of finetuning over 500 examples of MATH sampled from all levels. Even over a more difficult domain, we see significant improvements from multiagent finetuning that continue to self-improve.

| LLM | Methods | MMLU |
|---|---|---|
| LLaMA-3 (Dubey et al., 2024) | Base | $60.40 \pm 2.18$ |
| | Majority | $61.80 \pm 2.17$ |
| | Debate | $65.80 \pm 2.12$ |
| | Majority FT | $63.40 \pm 2.15$ |
| | Ours | $\mathbf{68.80} \pm 2.07$ |

Table 7: **MMLU Evaluation** We introduce an additional evaluation with the MMLU benchmark, finetuning on 500 MMLU examples and testing on 500 different MMLU examples. We find that our method performs better than other baselines.

We show results across all baselines in Table 6 and results across multiple rounds of finetuning in Figure 11. We see consistent improvement using LLaMA-3.

## H.2 MMLU

We add an additional comparison with MMLU to further establish thte improvement of our method on a task related to general factuality and reasoning instead of mathematics.

We finetune on 500 MMLU examples randomly sampled from all 57 subjects. We then evaluate on a different set of 500 randomly sampled examples.

We show results in Table 7. We see that our method can improve performance on a task related to factuality.

## H.3 ZERO-SHOT GENERALIZATION EVALUATION

We include a larger evaluation of zero-shot evaluation of our method in Figure 12, where we finetune on 500 MATH problems and test on 1000 GSM problems. We find that our method performs significantly better than all other baselines.

Furthermore, we test another setting to measure zero-shot performance by finetuning on the arithmetic dataset and evaluating on the GSM dataset. We finetune using 500 arithmetic problems and evaluate each method on 1000 GSM problems. See Figure 13. We find that our method also performs significantly better than all other baselines.

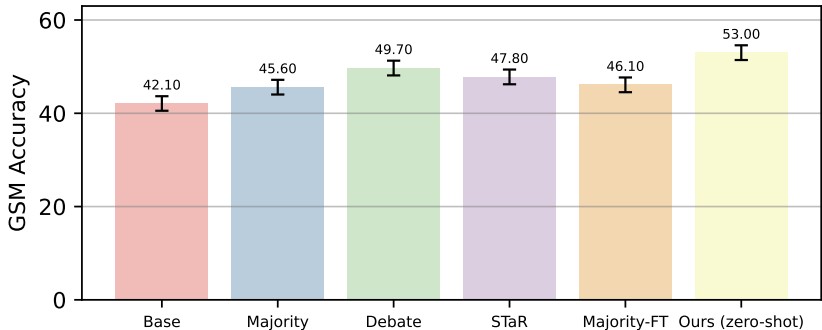

Figure 12: **Testing zero-shot generalization across 1000 GSM problems** We test the zero-shot capabilities of our method using models trained on the MATH dataset. We find that over 1000 problems of GSM, our method performs better than all baselines.

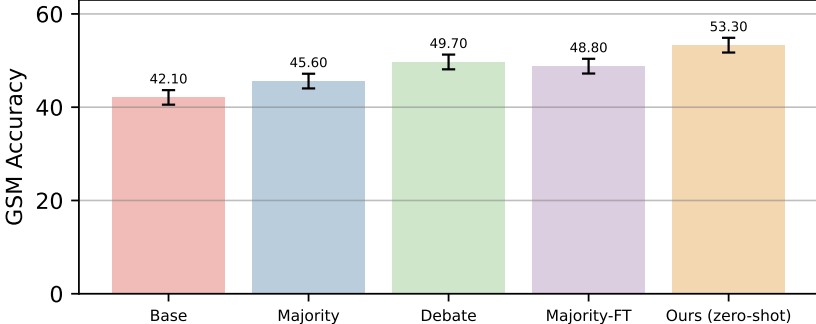

Figure 13: **Zero-shot generalization after arithmetic finetuning**. We evaluate the ability of our method to generalize after finetuning Mistral on the arithmetic task and evaluating on GSM. We find that this aids in GSM performance, even more than finetuning with MATH.

