# OpenReview forum: "Multiagent Finetuning: Self Improvement with Diverse Reasoning Chains"
_ICLR.cc/2025/Conference — ICLR 2025 Poster_

### Official Review · Reviewer_85F3 · 2024-11-03

**Soundness:** 3
**Presentation:** 2
**Contribution:** 3
**Rating:** 6
**Confidence:** 4

**Summary:**

This paper explores the fine-tuning of multiple LLM agents within the framework of multi-agent debate. The authors propose a method where multiple LLMs are fine-tuned and organized into generation and critic agents to facilitate multi-agent debate. Experiment results indicate that the proposed method outperforms baseline methods, and the authors further demonstrate that fine-tuning multiple LLMs helps preserve diversity compared to fine-tuning a single LLM.

**Strengths:**

1.	Jointly optimizing the LLM in the roles of generators and critics appears to be a robust method for enhancing the reasoning ability of LLMs.
2.	The work shows that finetuning multiple LLMs on independent datasets derived from multi-agent debate can preserve diversity, which is a critical challenge for LLM finetuning.
3.	The evaluation results show the strength of the proposed method.

**Weaknesses:**

1.	The title “Multiagent Finetuning of Language Models” may imply a broader scope than the paper addresses. Multi-agent applications of language models can indicate a much broader range of settings besides reasoning tasks and multi-agent debate, such as gaming and social simulation; however, this work focuses solely on multi-agent debate.
2.	The terms  “Single Agent” and “Multi Agent” is vague and unclear in this paper. For example, Sec 2.2 “Fine-tuning Single Agent”discusses scenarios involving multiple agents rather than a true single-agent setting.
3.	This work might not obey the standard training and evaluation procedure on GSM and MATH dataset, as only 500 examples are selected for training.

**Questions:**

1.	Line 77: How does the proposed approach “promotes diversification within the society of models”?
2.	What precisely does "Single-Agent" in "Fine-tuning Single Agent" refer to? Is it intended to indicate one generation agent or a single LLM? If the latter, a more fitting term might be “Fine-tuning Single LLM.”
3.	How is a fair comparison with baseline methods established?
4.	How does varying the number of agents affect the performance of the proposed method?
5.	It is pretty costly to train multiple LLMs, especially considering the inference-time compute and resources required to serve N LLMs. A straightforward possible strategy to avoid training multiple LLMs while also maintaining diversity is to include an unique identifier (e.g. an ID) or a special token in the input for each agent. How does this strategy compare to finetuning multiple LLMs?

---

> ### Author Response · Authors · 2024-11-23
> **Response to Reviewer 85F3 (Part 1)**
>
> > Q1: may imply a broader scope than the paper addresses. Multi-agent applications of language models can indicate a much broader range of settings besides reasoning tasks and multi-agent debate
>
> We have included an experiment in Appendix  D to address this, which shows our approach can be incorporated with LLM agents which are applied collaboratively rather than competitively as with debate. We find that multiagent finetuning improves performance over the collaborative settings as well and have updated the introduction with this point.
>
> > Q2: “Single Agent” and “Multi Agent” is vague and unclear in this paper.
>
> We will make the distinction more clearer. Single Agent refers to finetuning a single agent and using that single finetuned agent at inference time. The data generation method for that agent could use multiagent debate or some other simpler method for prompting like chain-of-thought. The result is only a single finetuned agent that we can apply complex prompting methods at inference time such as multiagent debate. To answer your question specifically, this refers to the equivalent of one generation agent or standard finetuning using multiagent debate.
>
> > Q3: This work might not obey the standard training and evaluation procedure on GSM and MATH dataset, as only 500 examples are selected for training.
>
> Thank you. Due to limited time, we have expanded the evaluation procedure on GSM to finetune on 1K examples and evaluate on 1K examples using LLaMA-3.
>
>
> | Method        | GSM Accuracy     |
> |---------------|------------------|
> | Base          | 70.50 $\pm$ 1.44 |
> | Majority      | 71.30 $\pm$ 1.43 |
> | Debate        | 77.80 $\pm$ 1.31 |
> | Majority FT   | 77.60 $\pm$ 1.31 |
> | Multiagent FT | 84.50 $\pm$ 1.17 |
>
> > Q4: Line 77: How does the proposed approach “promotes diversification within the society of models”?
>
> A key observation of multiagent debate and other multiagent prompting methods is that a single LLM can answer questions using multiple reasoning approaches that emerge from sampling on the probability distribution from the model. Given that the same base LLM can have different answers, the power of an approach like debate is the ability to generate diverse proposals, some of which lead to the correct answer, and integrate them by evaluating the correctness of approaches from other agents.
>
> Multiagent finetuning preserves and emphasizes this diversity by separating the finetuning across each agent. Finetuning leads to a collapse of internal diversity. However, by finetuning each agent separately on the dataset that the agents correctly generated in the previous, we identify reasoning chains that lead to the correct answer and preserve these reasoning chains in individual agents. This feedback loop promotes specialization within the society of models. As an agent finetunes on its own specific responses that led to a correct answer, it can become an expert on specific problems. This leads to stronger specialization over iterations of finetuning.  In a toy model mathematical model in Appendix Section G, we provide a simple illustration on how such mathematical specialization can happen.
>
> > Q5: How is a fair comparison with baseline methods established?
> Our baselines use different strategies that include (1) different prompting strategies over base, unfinetuned LLMs, (2) multiagent finetuning with simpler data generation methods than debate, and (3) finetuning methods like STaR. Base provides a comparison between all methods and using a single agent at inference time. Majority provides a comparison with a multiagent prompting approach where we simply take a majority vote over the agent responses. Similarly, Debate is the closest comparison to our approach since it uses a similar number of models at inference time. Finally, Majority FT and STaR are finetuning methods. Majority FT is a finetuning method that uses multiple agents, like our method. STaR is a state-of-the-art method for self-improvement.
>
> > Q6 How does varying the number of agents affect the performance of the proposed method?
>
> Appendix Section F includes an experiment with 5 agents. We find that more agents leads to improved results overall.

---

> ### Author Response · Authors · 2024-11-23
> **Response to Reviewer 85F3 (Part 2)**
>
> > Q7: It is pretty costly to train multiple LLMs, especially considering the inference-time compute and resources required to serve N LLMs. A straightforward possible strategy to avoid training multiple LLMs while also maintaining diversity is to include an unique identifier (e.g. an ID) or a special token in the input for each agent. How does this strategy compare to finetuning multiple LLMs?
>
> Thank you for this suggestion. We include an experiment where we avoid training multiple LLMs by feeding a simple unique ID at the beginning of each agent as a method for preserving diversity. Given a generation agent, we feed an ID of the form GEN1 where GEN refers to a generation agent and 1 refers to the first agent. Similarly, critic agents are fed IDs in the form CRIT1. We also include a small description of what generation and critic agents are meant to do at the beginning of the prompt. In general, we find this method comparable to debate. Results are shown in Appendix Section I.  We paste results here as well: https://ibb.co/yYY46CZ.

---

> ### Comment · Reviewer_85F3 · 2024-11-23
>
> Thank you for your feedback. Here I have some further questions.
>
> > The title “Multiagent Finetuning of Language Models” may imply a broader scope than the paper addresses
>
> As mentioned in the original review comment, “Multiagent Finetuning of Language Models” can refer to settings like web agents or gaming where the LLM does not just perform reasoning but also decision-making.
>
> > How is a fair comparison with baseline methods established?
>
> How many training tokens and how many rounds of debates are used in the fine-tuning process? A fair comparison should be ensured with a similar amount of training tokens or a similar number of reasoning trajectories.
>
> >  “Single Agent” and “Multi Agent” are vague and unclear in this paper.
>
> The notions of "agent" and "LLM" should be used rigorously in the context of the paper where multiple LLMs are involved and multiple agent types and debate agents are involved. Multiple agents can be supported by different LLMs or a single LLM. The notion of "single-agent" was confusing for me since the authors use a single agent for multiagent debate, though the actual case is using a single fine-tuned LLM for multiagent debate.
>
> > "How does varying the number of agents affect the performance of the proposed method?"
>
> Additional experiments on the number of agents resolve my concerns regarding this question. Is there any discussion on the benefits and limitations of the approach if significantly larger amount of agents are used?

---

> ### Author Response · Authors · 2024-11-24
> **Follow-up Response to Reviewer 85F3**
>
> We thank the reviewer for reading through our response! We address the additional comments below.
>
> > Q1:  “Multiagent Finetuning of Language Models” can refer to settings like web agents or gaming where the LLM does not just perform reasoning but also decision-making.
>
> Thank you for the feedback. We recognize that the title may be too broad. We will change the title to convey that our focus is primarily on self-improvement. Our new title is Self Improvement in Language Models through Multiagent Finetuning. We’re also happy to further change the title of the paper if the reviewers so desires.
>
> > Q2: How many training tokens and how many rounds of debates are used in the fine-tuning process?
>
> For our method, we always used 2 rounds of debate. This is the same as the Debate baseline, which does not use finetuning.
> We compare the number of training tokens for generation agents and critic agents in our method in comparison to the number of training tokens used for Majority-FT. We evaluate this on the MATH dataset with Mistral-7B. For multiagent finetuning, a generation agent is finetuned on 55889 tokens and a critic agent is finetuned on 119645 tokens. In comparison, we use 353814 tokens with Majority FT. Majority-FT uses a majority vote over final responses and collects all responses that match the majority voted answer. Our method only selects one response per input example. Furthermore, generation agents are only finetuned on examples that have a final answer in the first round which matches the majority voted solution in the second round. This reduces the number of selected examples further in comparison to Majority-FT. So we finetune on significantly more tokens with Majority-FT in comparison with our method. We believe this shows that our comparison with other finetuning baselines is fair since we finetune on fewer tokens in our approach (with the total aggregated number of tokens across multiagent finetuned models similar to that of Majority-FT).
>
> > Q3: The notions of "agent" and "LLM" should be used rigorously in the context of the paper where multiple LLMs are involved and multiple agent types and debate agents are involved. Multiple agents can be supported by different LLMs or a single LLM.
>
> We apologize for the confusion between “model” and “agent”. We have significantly revised the method section including section titles to make the distinction clear that a “LLM model” are the parameters that are trained, and an agent is constructed from the model. We have highlighted all changes in blue.
>
> > Q4: Is there any discussion on the benefits and limitations of the approach if significantly larger amount of agents are used?
>
> This is a great point! Reviewer Tw8m asked a similar question as well. We indeed have an experiment illustrating how the approach works well with 5 agents in Appendix Section F, where additional agents lead to significant improvement across all tasks, and significantly outperforms all baselines, illustrating how we can use an even larger number of agents in our approach. Theoretically, Appendix Section G shows that more agents allow for better coverage of different specialization skills, outside of three used as part of the original analysis. In addition, prior work [1] has shown significant benefit from incorporating more agents in inference time approaches like multiagent debate.
>
> However, a limitation of using more agents is added compute and memory complexity. For example, we had hardware limitations that prevented us from experimenting with additional agents on open-source models, as each additional model required additional GPU time to train. There are some initial ways of overcoming these limitations. A promising venture could include some of the smaller LLMs that have 1-2B parameters such as LLaMA-3.2. Memory optimization methods like quantization could also be applied to allow for more agents to be loaded in memory.
>
> [1] Du et. al. Improving Factuality and Reasoning in Language Models through Multiagent Debate, ICML 2024.

---

> > ### Comment · Reviewer_85F3 · 2024-11-24
> >
> > Thanks for your update. I believe the authors' responses have resolved my concerns. I have raised my score accordingly.

---

### Official Review · Reviewer_TeFj · 2024-11-03

**Soundness:** 3
**Presentation:** 3
**Contribution:** 3
**Rating:** 8
**Confidence:** 3

**Summary:**

This paper proposes a method to address a key limitation of LLMs: their dependency on static datasets, which limits their ability to self-improve. The authors propose a multiagent approach for LLM finetuning that enables iterative self-improvement through interactions with other models. By using distinct Actor (generation) and Critic (evaluation) agents, their approach improves feedback quality and response diversity, which results in better responses over iterations of finetuning. The authors empirically demonstrate their approach's effectiveness across 3 reasoning benchmarks (Arithmetic, MATH, GSM), showing performance gains of approximately 1-15% across 4 baselines. Additionally, the finetuned agents may demonstrate zero-shot generalization from MATH to the other two benchmarks.

**Strengths:**

- Significance: This paper presents a promising approach to LLM self-improvement and could offer a valuable contribution.
- Clarity: Most of the Figures in the paper are clear and the paper is generally well-written.

**Weaknesses:**

There are several comments I would like the authors to address to make some details clearer and the paper more complete.

**Major comments**
1. Role Specialization: The paper introduces distinct roles for models (generation agents and critic agents). However, it would be helpful to clarify the specific objectives each role optimizes. Additionally, I suggest emphasizing that only two roles are used in this paper (generation and critic) to avoid confusion.
2. Zero-shot Generalization: In Section 4.3 The authors claim zero-shot generalization on held-out benchmarks. I find the claims made in Section 4.3 not very convincing for three reasons:
- (a) The authors only evaluate on 100 random samples. Could the authors elaborate on why this specific sample size was chosen, and why not more?  If it is computationally possible to evaluate more samples (e.g. 1000), this would make the evidence for zero-shot transfer more convincing.
- (b) In Figure 5, the standard errors for performance (accuracy) are absent; could you please report these like was done in the Table 2
- (c) The choice of finetuning on MATH and evaluating on the GSM dataset is unclear; could you explain the rationale? Why not report all cross-dataset generalizations, such as MATH → GSM or Arithmetic → GSM? If the results are consistent across all cross-evaluations, this would strengthen the findings.
3. Computational Trade-offs: The limitations section mentions increased computational demands but does not quantify these trade-offs. Details on the additional costs or time requirements would help to evaluate the compute/performance balance.
4. Limitations Discussion: The current discussion of limitations is very short. Future researchers building on this work would benefit from more detailed insights into any bottlenecks or constraints of the approach. Based on these more detailed suggestions for future research directions would be valuable.
5. Response diversity: Looking at the slope, Figure 3 seems to indicate that all the benefits could be coming from the initial improvement in diversity. It seems important to mention the initial diversity for each ablation.
6. Standard error: How exactly is the standard error in Table 2 computed?

**Minor comments** (that did not affect my score)
- Abstract: The sentence“ A set of language models are initialized from the same base model and then are specialized by independently updating each model using data generated by the model under multiagent interaction with other models” is somewhat lengthy and could be simplified for readability.
- Citation Formatting: Throughout the paper, `\cite{}` is used in places where `\citep{}` might be more appropriate (e.g., lines 137, 244, 448).
- Figure: In Figure 4, what are the multiple dots in each box for each method? I’m assuming that their order is also the order of performance across iteration but it’s not obvious.

**Questions:**

See my questions in the "Major comments"

---

> ### Author Response · Authors · 2024-11-23
> **Response for Reviewer TeFj (Part 1)**
>
> We thank the reviewer for their thoughtful review. We address the comments below.
>
> > Q1: However, it would be helpful to clarify the specific objectives each role optimizes. Additionally, I suggest emphasizing that only two roles are used in this paper (generation and critic) to avoid confusion.
>
> We restate the roles of our agents and their optimization procedure precisely here.
>
> Generation Agent: The role of a generation agent is to generate accurate responses to input questions. These should rely on diverse reasoning chains to promote diversity. The finetuning procedure is as follows. Each generator agent is trained on a dataset curated from its response from the first round. For example, the $n$-th critic agent uses the text from $\{y_{1, n}\}$. We only keep data where the correct solution is achieved i.e. $y_{1, n} = \hat{y}$ is achieved as shown in Figure 2 in the main paper.
>
> Critic Agent: The role of a critic agent is to provide accurate critiques to responses from other agents and use these responses to provide an updated answer. The finetuning procedure is as follows. Each critic agent is trained on a dataset curated from its own responses and the corresponding responses from previous rounds. For example, the $n$-th critic agent uses the text from $\{y_{1,n} \text{ to } y_{M,n}\}$. The critic agent might generate either correct or incorrect solutions. We only keep the data where the correct solution $y_{M,n} = \hat{y}$ is achieved, as shown in Figure 2 in the main paper. For these data points, if the response of the corresponding generator in the first round $y_{1,n}$ is wrong, we classify such data as negative data. Including this data allows the critic agent to develop the ability to correct answers even when the initial response is wrong. The remaining data, where both the generator's and the critic's responses are correct, are classified as positive data. We sample positive and negative data from these two sets and  mix them together to train the critic agent.
>
> > Q2:  find the claims made in Section 4.3 not very convincing
>
> We address the three objections point by point.
>
> (a) We agree! We re-run an extended evaluation on 1000 data points for all reported numbers. We find that our method significantly outperforms all baselines on 1000 held out data points when finetuning on a different dataset. We find that this performs significantly better than all other baselines. We paste results here: https://ibb.co/JKpZ7WJ.
>
> (b) We apologize. This should have been included in the final result. We have added error bars to the figure to analyze for significance, please see Figure 11 in the new version of the paper. Our method has significant improvement when finetuning on MATH and testing on GSM.
>
> (c) We include a new setting where finetune on the Arithmetic and test on GSM and report this in Appendix Section L. We find that this also performs significantly better than all other baselines. We past the result here: https://ibb.co/WnpGh4Z.
>
> > Q3: The limitations section mentions increased computational demands but does not quantify these trade-offs.
>
> This is a fair point that we should have made clear. To run experiments we used a combination of eight H100 GPUs and four 40GB A100 GPUs. Running multiagent finetuning was difficult due to memory usage from loading in 6 finetuned models when using open source models. In general, this takes a total of 120 GB of GPU memory for Phi-3 and about 240 GB of memory for larger models like Mistral and LLaMA-3 if you load all six finetuned LLMs at once. In general, we found that inference time for 500 problems took between 12-16 hours depending on the size of the model when using multiple GPUs. When using one GPU but changing which agent is on GPU, inference time could take 24 hours. For open-source models, code was written in pytorch but designing models in JAX/FLAX may lead to further improvements in runtime. We expand our limitations section with this discussion.
>
> > Q5: Initial improvements in diversity
> We compare the diversity measures after one iteration of finetuning for Mistral using responses from the ablations in Table 2.
>
> | Method                    | Diversity: Embedding Dissimilarity |
> |---------------------------|------------------------------------|
> | Multiagent FT (ours)      | 0.457                              |
> | Multiagent FT w/o summary | 0.453                              |
> | Multiagent FT w/o critic  | 0.383                              |
> | Single Agent FT           | 0.299                              |
>
>
> We find that methods using multiagent finetuning generally has more diverse responses than generating responses using a single finetuned agent. We can attribute the close diversity for Multiagent FT and Multiagent FT w/o summary to summarization having little effect on diversity but instead on incorporating information and accuracy.

---

> > ### Author Response · Authors · 2024-11-23
> > **Response to Reviewer TeFj (Part 2)**
> >
> > > Q6: Standard error computation
> >
> > We compute all standard errors in the paper over the test samples by taking the standard deviation over the accuracies of each individual sample and dividing this by the square root of the total number of samples.
> >
> > > Q7: somewhat lengthy and could be simplified for readability.
> >
> > Thank you. We have updated this sentence.
> >
> > > Q8: Citation Formatting
> >
> > We have updated the citation. Thank you for pointing this out.

---

> > > ### Comment · Reviewer_TeFj · 2024-11-23
> > > **Response**
> > >
> > > Thank you for addressing my concerns. Unfortunately, I can't view the images, could you please check them?

---

> > > > ### Author Response · Authors · 2024-11-23
> > > > **Response**
> > > >
> > > > Our apologies! It seems our image hosting website went down. The images have reuploaded and should be viewable now.

---

> > > > > ### Comment · Reviewer_TeFj · 2024-11-24
> > > > > **Response**
> > > > >
> > > > > Thank you! The authors have addressed my comments, and I increased my score accordingly. I will vote for acceptance

---

### Official Review · Reviewer_Tw8m · 2024-11-04

**Soundness:** 2
**Presentation:** 3
**Contribution:** 3
**Rating:** 6
**Confidence:** 4

**Summary:**

This paper contributes to the field of self-improvement fine-tuning for LLMs by proposing a multi-agent cooperation approach. It replicates an LLM into multiple generation agents and corresponding critic agents. A multi-agent debate architecture is utilized to generate label responses for generation agents and critic agents. Each agent is then fine-tuned using its unique generated dataset. During the inference phase, the final result is generated following the multi-agent debate. The method demonstrates superiority over the baselines in a series of math-related language reasoning tasks.

**Strengths:**

1. The paper is easy to follow and the content is well-organized.
2. The paper proposes a method for agent self-improvement fine-tuning based on multi-agent collaboration, allowing for multiple rounds of self-improvement fine-tuning, which could be a promising approach.

**Weaknesses:**

My primary concerns with this paper are centered around the experimental section.

(Major)The first concern is regarding the selection of experimental datasets. The paper exclusively uses mathematical language reasoning tasks, and each task is not particularly challenging. Arithmetic is limited to arithmetic operations, GSM corresponds only to Grade School level difficulty, and MATH selects only the first three levels. If the tasks are not challenging enough, it may lead to questioning the need for incorporating multiple LLMs when a single LLM might be adequate. What’s more, conducting generalization experiments solely within mathematical datasets may not demonstrate the unique advantages in terms of generalizability, given the highly similar nature of the tasks. Including a graduate-level mathematical reasoning set or challenging datasets from other fields would make the experiments more convincing.

(Major)The second concern is that the performance improvement on datasets of modest difficulty is not significant, especially considering the introduction of multiple large models collaborating. It may be worthwhile to quantify other metrics beyond answer accuracy, for example, the KL divergence of each LLM from the original LLM distribution? Introducing multiple large models might allow agents to achieve better results without significant parameter changes.

*Overall, I believe the idea of the paper is commendable. However, as a practice-oriented paper lacking in theoretical explanation, its experimental design is insufficient. Without addressing these major concerns in the experiments, I'm afraid this is the highest score I can give.*

（Minor）The diversity of models: When reading the sentence " each model to capture parts of a task of interest.", I am very excited. However, the authors do not take any measures during the implementation to activate the heterogeneous characteristics among the generative agents. In other words, diversity is only brought about by the quantity of agents. The diversity cannot be theoretically guaranteed, nor can it be conceptualized and defined for specific agents after it appears. If the authors could enable different agents to collaboratively complete a complex task and truly capture the sub-tasks they are interested in or excel at, it might greatly enhance the contribution of this paper.

**Questions:**

In addition to my primary concerns, there are also the following questions:
1. In this paper, is each individual agent fine-tuned using the SFT (a combination of query and label response data) approach? If yes, can this method be integrated with DPO (a combination of query and human feedback preferences data) and PPO (a combination of query and reward signal data) approaches? Please provide a detailed discussion.
2. How is the Combination of datasets implemented in line 25 of Algorithm 1 pseudocode? What does the hyperparameter denote? The authors need to clearly explain how the two types of data are used respectively in the training of the critic agent and what the role of the weight hyperparameter is.
3. The abstract mentions “multiagent society of language models”, yet the experiments only utilized 3-5 agents (and the improvement with 5 agents does not seem significant). How do the authors view the relationship between the number of agents and performance improvement, as well as the issue of number versus resource consumption?
4. The selection of baselines. From the setup of the baselines, this work seems more like an “A+B+C” approach. However, would it be fair to compare “B+C” with B alone, even if the A module is ablated?

---

> ### Author Response · Authors · 2024-11-23
> **Response for Reviewer Tw8m (Part 1)**
>
> We thank the reviewer for their thoughtful and constructive review. We address specific points below.
>
>
> > Q1: The paper exclusively uses mathematical language reasoning tasks, and each task is not particularly challenging.
>
> Thank you! We include a more challenging setting by using **all levels of the MATH dataset** instead of levels 1-3, evaluating over LLaMA-3 (8B). We find that multiagent finetuning has significantly better performance than all baselines and has continuous improvement across iterations of finetuning. We compare with baselines here: https://ibb.co/fqppHjW and over iterations of finetuning here: https://ibb.co/qgJ0WwC.
>
> We also include a result on a dataset related to **factuality, MMLU**. We find that multiagent finetuning has significantly better performance, achieving 68.80%. See results here: https://ibb.co/7r722kp.
>
> Our proposed method outperforms baseline approaches not only on the more challenging math tasks but also on tasks beyond mathematics, such as those requiring factual accuracy.
>
>
> > Q2:  The performance is not significant, especially considering the introduction of multiple large models collaborating. It may be worthwhile to quantify other metrics beyond answer accuracy, for example, the KL divergence of each LLM from the original LLM distribution?
>
> We thank the reviewer for their valuable feedback. We believe that our method does demonstrate a significant improvement over the baselines in terms of answer accuracy as illustrated by our statistically significant results in Table 1 and other tables in paper.
>
> As suggested by the reviewer, we have added an additional experiment computing the KL divergence of each LLM from the original LLM distribution. We have included results in the new Figure 10 in the paper and also an attach an image of the results here https://ibb.co/10FDvbW. Our approach obtains a significantly higher KL divergence from the original LLM distribution than single model fine-tuning. In addition, we have also added a new Figure 13 where we measure the likelihood of generated responses from one agent under other multiagent fine-tuned agents. In this figure, we find additional evidence of diversity in the agents over rounds of finetuning. This is shown here as well: https://ibb.co/FscDc97.
>
> In addition, we would like to note that in the original submission, we already included evaluations of diversity on a broader range of metrics. For example, in Figure 3 we include embedding distance,  in Figure 7 in Appendix Section C.2, we analyzed the KL divergence between our proposed method and the baseline method that uses single-agent fine-tuning. Additionally, in Figure 6 of Appendix Section C.1, we evaluated the consensus of the generated results. Our method demonstrated greater consensus and diversity compared to the baseline across all these axes.
>
> > Q3: In other words, diversity is only brought about by the quantity of agents. The diversity cannot be theoretically guaranteed, nor can it be conceptualized and defined for specific agents after it appears.
>
> One way to conceptualize the findings in this work is to think of a toy setting where we apply an LLM on a synthetic task with subtasks -- A, B, C -- with an initial distribution of 33/33/33 among the subtasks. We can activate the heterogeneous characteristics among agents in this framework where the agents are finetuned on their own outputs, naturally specializing them in tasks the agents excel at.
>
> Consider a scenario with 3 agents. If we expose each agent to an individual subtask of specialization, this guarantees agent specialization. This is because we only finetune each agent on answers it gets correct using multiagent debate. Over iterations of finetuning, the process leads to self-improvement where the agent becomes specialized in task A, B, C. Theoretically, this process should converge to a state where each agent exhibits 100% on the respective subtasks. This shows that diversity and task-specific heterogeneity are naturally induced without additional intervention. We provide a mathematical derivation of this phenomenon in Appendix Section G.
>
> One question is how subtasks manifest. This happens from internal model specialization. LLMs are probabilistic. They can respond to questions in different ways using different reasoning chains that are derived based on sampling. The ability to specialize is useful for multiagent inference methods like debate where different reasoning chains can be used to find the answer to a question. Our finetuning method encourages individual agents to concretely discover subtasks -- A, B, C -- and finetune to enhance specialization.

---

> ### Author Response · Authors · 2024-11-23
> **Response for Reviewer Tw8m (Part 2)**
>
> > Q4: If yes, can this method be integrated with DPO (a combination of query and human feedback preferences data) and PPO (a combination of query and reward signal data) approaches? Please provide a detailed discussion.
>
> Yes, all agents are finetuned using SFT. This means that all agents could also be finetuned using DPO and PPO, as long as there are preferences between responses. One way to incorporate DPO and PPO would be to have critics generate preferences between responses or assign rewards in order to guide optimization. For example, in DPO, critic agents would assign preferences and these preferences can be leveraged to align generation agents’ outputs with the preferred response, optimizing for higher-quality outputs. PPO would have a similar mechanism, where reward signals from critic agents can be incorporated, allowing generation agents to iteratively refine their responses through reward feedback. This could allow the feedback to go beyond binary correctness and use more subjective metrics.
>
> There are challenges with integrating DPO or PPO. Preferences or reward signals from critics may be noisier and less robust in comparison to explicit correctness labels used in SFT. This could lead to instability during training, which has been observed in prior work with PPO [1]. PPO, in particular, may require additional tuning to manage noisy signals increasing computational complexity compared to SFT. SFT offers simplicity, reliability, and efficiency, but DPO and PPO are useful for scenarios where labels aren’t available or when we want to optimize for more subjective metrics beyond correctness.
>
> > Q5: How is the Combination of datasets implemented in line 25 of Algorithm 1 pseudocode? What does the hyperparameter denote?
>
> The hyperparameter term "w" refers to a weighting parameter used for sampling data from two different sets. In this context, "w" represents the proportion of data sampled from the first set, while "(1 - w)" represents the proportion of data sampled from the second set. This has been updated in the text.
>
> > Q6: How do the authors view the relationship between the number of agents and performance improvement, as well as the issue of number versus resource consumption?
>
> We believe that incorporating more agents will lead to better performance. Empirically, this has been noticed in inference-time approaches like multi-agent debate where more agents leads to stronger results. Intuitively, our setting will also benefit from additional agents. More agents will lead to more specialization through finetuning as we cover more reasoning chains stored in the model. However, running experiments with many agents is challenging as it is very computationally expensive to train 5 or more separate models. We find in our experiments that there are significant result even with 3 agents and believe this trend with more agents.
>
> In terms of performance of 5 agents reported in Table 4 of the paper, we note that our approach with multiagent finetuning does lead to a large boost in performance compared to baseline methods. The performance gains may look smaller than those than Table 1, but this is because we only report comparisons with the strongest baselines in Table 1, and because all methods already get much higher performance with more agents, giving less room for additional improvements.
>
> > Q7: The selection of baselines. From the setup of the baselines, this work seems more like an “A+B+C” approach.
>
> We're not completely sure what the reviewer means by a "A+B+C" approach. Our approach gives a method in which we can directly specialize a set of language models so that they can coordinate together to solve complex reasoning tasks.
> There isn't really a specific "A" + "B" + "C" set-up in this approach (unless the reviewer refers to the individual components of multiagent debate) where we can directly swap a baseline approach for one component and maintain the other components of our approach (since other methods focus solely on either self-improvement or test-time inference). If the reviewer refers to the individual components of multiagent debate, we actually investigate the effect of individual components of our approach such as multiple models, debate, and separate verifier / critic models in Table 2 of the paper.
>
> [1] Rui Zheng et. al. Secrets of rlhf in large language models part i: Ppo. arXiv, 2023.

---

> ### Comment · Reviewer_Tw8m · 2024-11-25
>
> Thanks for your response, which has addressed most of my concerns. I now have primary questions regarding the diversity among agents. Firstly, I am curious about the mechanisms through which diversity among agents is achieved. The paper does not seem to have an explicit mechanism for the definition and assignment of agents’ roles. I would appreciate it if the authors could provide a specific example to illustrate this. Secondly, I would like to delve deeper into the origin of this diversity. For instance, in the case of five agents, does their diversity stem just from the increased number of agents, or is it a result of collaboration among the agents?
> Considering my initial rating was between 4 and 5 (the highest possible at the time), I will now add one point, without changing my level of confidence, making my temporary score a 6.

---

> > ### Author Response · Authors · 2024-11-26
> >
> > We thank the reviewer for their response. We hope to provide additional discussion here. Please let us know if you have any additional questions which we can answer in subsequent responses.
> >
> > > Q1: paper does not seem to have an explicit mechanism for the definition and assignment of agents’ roles. I would appreciate it if the authors could provide a specific example to illustrate this
> >
> > Thank you for this comment. The definition and assignment of each agents’ role is done implicitly through recursively training an agent on its own responses (which leads to specialization as seen in the newly added mathematical model in Appendix G).  We agree that an example would be much more useful to show the kind of specialization models have after finetuning. We include an example of generated responses for one problem from GSM in a simpler setting where we have two generation agents and two critic agents which are based on the Mistral. We show the responses from the initial, base models (https://i.postimg.cc/RV6kJMfP/image.png) and from the finetuned models (https://i.postimg.cc/xTwWKW39/image.png). We believe this is a good representation of the kind of specialization we aim for in this paper with a simple enough example. We can observe that before finetuning, one agent uses algebraic thinking to get the correct answer but another uses a random guess-and-check approach. One of the critic agents completely disregards the solutions from other agents and provides their own answer, completely ignoring its role in analyzing responses.
> >
> > However, after one iteration of finetuning the models, we notice that while both are using algebraic approaches, one is using a direct linear equation while the other is using a system of equations. One of the generation agents still gets the answer incorrect but both critic agents are able to get the correct answer but use different reasons. The first critic agent seems to be analyzing for simpler approaches while the second critic agent does a step-by-step evaluation of the result. This illustrates how specialization manifests through finetuning to incorporate different approaches to solving a math problem for example. Note that we have not explicitly told agents to specialize in this way -- this division is unsupervised and manifests from the fact that by random chance each agent gets different problems correct with different chains of reasoning. Thus, since agents are trained on heterogenous data, they exhibit different strengths and reasoning chains, with this difference accentuated over repeated rounds of finetuning (Appendix G).
> >
> > > Q2: does their diversity stem just from the increased number of agents, or is it a result of collaboration among the agents?
> >
> > This is a great question and we introduce a new ablation to address this. In this ablation, we attempt to assess the diversity of response when removing interaction among agents. We construct a no-interaction baseline, where we measure the diversity of final responses using multiagent finetuned models,  but where all generations are from 1 fixed generator (removing the interaction between generation agents and critic agents).
> >
> > We show the results here: https://i.postimg.cc/D0Kq2Dj6/image.png. Overall, we observe that interaction is important for diversity: the no interaction method has substantially lower diversity over final generation responses compared to multiagent FT models under the negative log-likelihood diversity metric. Simultaneously, we still see that the no interaction baseline still outperforms a single-agent finetuned model, indicating that an increased number of agents also leads to more diversity.

---

### Author Response · Authors · 2024-11-23
**General Response**

We thank the reviewers for their constructive reviews. We are glad reviewers found the paper well-written and easy to follow (Tw8m, TeFj), found the approach promising in preserving diversity and improving performance (Tw8m, TeFj, 85F3), and found the evaluation robust (85F3). We address some common feedback here as well as provide an overview on what we have updated in the paper.

* More difficult tasks and datasets; larger scale evaluation (Tw8m, 85F3, TeFj): Reviewers requested further evaluation of multiagent finetuning on more difficult tasks and larger evaluations. We cover some additional experiments we included.
    * MATH Levels 1-5: Instead of evaluating on MATH examples from the first three levels, we evaluate on  examples from all five levels. We find that multiagent finetuning performs significantly better than all baselines in this case. This is included in Appendix Section H and results are visualized here: https://ibb.co/fqppHjW.
    * MMLU: We add an additional comparison with baselines with MMLU to further address concerns that our evaluation is only on math-related datasets. We include this as Appendix Section K and are visualized here: https://ibb.co/7r722kp.
    * Zero-Shot evaluation (TeFj): In the paper, we evaluate the zero-shot generalization capabilities of multiagent finetuning over 100 examples, transferring abilities gained from MATH to GSM. We evaluate 1000 examples to further assess capabilities on held-out generalization. We also assess whether we can train on arithmetic and transfer capabilities to GSM over 1000 examples. This is shown in Appendix Section L.
* Fairness of baselines (Tw8m, 85F3): Two reviewers asked about whether the baselines were fairly established for comparison. Our goal was to use baselines that use methods used for evaluating LLMs currently. Our baselines use different strategies that include (1) different prompting strategies over base, unfinetuned LLMs, (2) multiagent finetuning with simpler data generation methods than debate, and (3) finetuning methods like STaR.
* Limitations (TeFj): We add a further discussion of limitations to the paper.
* New diversity metric, likelihoods: We added a new diversity metric for measuring response diversity among our finetuned agents based on likelihood. In this metric, we take responses from two of our critic agents and feed them to a third held-out critic agent at a particular iteration. We measure the negative log-likelihood (NLL) of these responses. The same approach is used for our Single-Agent FT ablation but over general finetuned agents instead of critic agents. As agents diversify, we should expect NLL to increase across iterations of finetuning. This is exactly what we find as seen in Appendix Section M and we include the result here:  https://ibb.co/FscDc97. Our method increases the diversity of model responses while improving accuracy.
* Writing changes: We summarize writing changes here and all changes in the paper are highlighted in blue.
    * Section 2: We make the language clearer around generation agents and critic agents and clarify notation on the parameter “w”.  We make the distinction between single agent and multiple agents clearer.
    *  Appendix Section G: We include a new appendix section to discuss the theoretical intuition of our approach which includes the ideas discussed here.
    * We have fixed smaller typos with our citations and other writing typos. We thank reviewers for finding these.

---

### Meta-Review · Area_Chair_f3sd · 2024-12-20

**Metareview:**

This paper presents a novel and effective approach to enhancing LLM reasoning through jointly optimizing LLMs as generators and critics.  The reviewers appreciate the robustness demonstrated by this method, particularly its ability to preserve diversity during finetuning through independent datasets derived from multi-agent debate. The evaluation results convincingly support the effectiveness of the proposed technique.  While there are some concerns regarding the clarity of terminology and scope implied by the title, as well as the potential deviation from standard dataset usage, these issues do not detract from the core contribution of the work. The strengths of this paper, namely its innovative methodology and strong results, significantly outweigh the weaknesses. The authors are encouraged to revise the title and clarify terminology for the final version. Further clarification on the dataset usage would also be beneficial.  Overall, this paper offers a valuable contribution to the field of LLM reasoning and is well-suited for publication.

**Additional Comments On Reviewer Discussion:**

nothing outstanding.

---

### Decision · Program_Chairs · 2025-01-22

Accept (Poster)